# Reciprocal causation mixture model for robust Mendelian randomization analysis using genome-scale summary data

Zipeng Liu[1,2,3,7], Yiming Qin[1,2,3,7], Tian Wu[1], Justin D. Tubbs[1], Larry Baum[1,2,3], Timothy Shin Heng Mak[3], Miaoxin Li[1,4,5] ✉, Yan Dora Zhang[3,6] ✉ & Pak Chung Sham[1,2,3] ✉

Mendelian randomization using GWAS summary statistics has become a popular method to infer causal relationships across complex diseases. However, the widespread pleiotropy observed in GWAS has made the selection of valid instrumental variables problematic, leading to possible violations of Mendelian randomization assumptions and thus potentially invalid inferences concerning causation. Furthermore, current MR methods can examine causation in only one direction, so that two separate analyses are required for bi-directional analysis. In this study, we propose a ststistical framework, MRCI (Mixture model Reciprocal Causation Inference), to estimate reciprocal causation between two phenotypes simultaneously using the genome-scale summary statistics of the two phenotypes and reference linkage disequilibrium information. Simulation studies, including strong correlated pleiotropy, showed that MRCI obtained nearly unbiased estimates of causation in both directions, and correct Type I error rates under the null hypothesis. In applications to real GWAS data, MRCI detected significant bi-directional and uni-directional causal influences between common diseases and putative risk factors.

The advent of genome-wide association studies (GWASs) has confirmed widespread genetic correlations among numerous complex diseases and traits[1]. Such correlations may represent pleiotropic genetic effects on multiple phenotypes, or causal relationships between phenotypes[2]. The analysis of GWAS data may help to identify causal relationships between phenotypes, contributing to our understanding of disease etiology. In recent years, causal modeling using Mendelian randomization (MR) has been widely applied to GWAS summary data[3,4]. MR uses genetic variants, typically single nucleotide polymorphisms (SNPs), as instrumental variables (IVs)

which, to be valid, should be (1) associated with the exposure (the relevance assumption), (2) not associated with confounders of exposure and outcome (the independence assumption), and (3) only associated with outcome through exposure (the exclusion restriction assumption)[5]. Basically, MR estimates the causal effect of an exposure on an outcome by the ratio of the effect of a genetic variant on the outcome to the effect of the same genetic variant on the exposure. In practice, due to the high polygenicity of complex traits, multiple SNPs may be necessary to increase the power and robustness of MR[6].

[1]Department of Psychiatry, Li Ka Shing Faculty of Medicine, The University of Hong Kong, Hong Kong SAR, China. [2]State Key Laboratory of Brain and Cognitive Sciences, The University of Hong Kong, Hong Kong SAR, China. [3]Centre for PanorOmic Sciences, Li Ka Shing Faculty of Medicine, The University of Hong Kong, Hong Kong SAR, China. [4]Zhongshan School of Medicine, Center for Precision Medicine, Sun Yat-sen University, Guangzhou, China. [5]Key Laboratory of Tropical Disease Control (SYSU), Ministry of Education, Guangzhou, China. [6]Department of Statistics & Actuarial Science, Faculty of Science, The University of Hong Kong, Hong Kong SAR, China. [7]These authors contributed equally: Zipeng Liu, Yiming Qin. ✉e-mail: limiaoxin@mail.sysu.edu.cn; doraz@hku.hk; pcsham@hku.hk

However, using multiple SNPs as IVs also increases the chance of horizontal pleiotropy for some SNPs[7], which would violate the assumption for methods like inverse-variance weights (IVW)[4]. MR-Egger[8] and MR-PRESSO[9] address the issue of horizontal pleiotropy under the InSIDE (instrument strength independent of direct effect) assumption, while Weighted Median[10] and Weighted Mode[11] are robust to horizontal pleiotropy provided that there is an adequate proportion of valid IVs in estimation. More recent methods such as MRMix[12] assume a normal-mixture model to consider horizontal pleiotropic effects. While these methods address the violation of the MR assumptions by either removing pleiotropic SNPs or explicitly modeling pleiotropic effects, they still involve the selection of independent IVs, which may exclude the majority of SNPs, with consequent loss of information. One recent advance is to account for correlated pleiotropy by estimating the nuisance parameters of a mixture model from randomly selected genome-wide summary data. However, this method still requires a set of linkage disequilibrium (LD) pruned variants to calculate posterior distributions to fit the causation model[13]. As with other existing methods, inference of the causal relationship in the reverse direction requires a separate analysis with a different set of IVs[7,14], which may not always be a nontrivial process.

In recent years, causal modeling based on MR approaches has been applied to investigate the inter-relationships between a wide range of phenotypes for which summary GWAS data are available[15]. With such widespread usage and the proneness of MR methodologies to violations of model assumptions, there is a risk of false positive findings from such studies. False positive causal inferences may be particularly likely when most causal SNPs are pleiotropic so that few SNPs can serve as valid IVs[16].

In this article, we propose a statistical method called mixture model reciprocal causal inference (MRCI) to infer the causal paths simultaneously in both directions between two phenotypes. Our method uses GWAS summary statistics of all available SNPs for the two phenotypes, together with reference LD information on the SNPs. We consider SNPs to fall into four mutually exclusive effect categories (trait-specific, pleiotropic, and null SNPs) and construct a composite likelihood function that takes into account the LD among SNPs. Compared with existing MR approaches our method tests for reciprocal causation without the selection of IVs, thus making full use of genetic information while explicitly modeling pleiotropy. In particular, MRCI is robust even in situations where most causal variants are pleiotropic. We applied MRCI to cardiovascular and metabolic disorders and some of their putative risk factors using public GWAS summary data, with results that provided insights into the causal relationship between several pairs of phenotypes.

## Results
### Overview of MRCI
All phenotypes and additively coded genotypes were assumed to have been standardized to have unit variance. In the full model, SNPs were assumed to fall into four mutually exclusive components: trait-specific $(G_1, G_2)$, pleiotropic $(G_C)$ and null SNPs $(G_0)$, with mixing proportions $\pi_1, \pi_2, \pi_c$ and $\pi_0$, respectively. SNPs in the $G_1$ and $G_2$ components have direct effect sizes of $\gamma_1$ and $\gamma_2$ on phenotype $Y_1$ and $Y_2$ respectively, while $G_C$ SNPs have direct effect sizes of $\gamma_{C1}$ and $\gamma_{C2}$ with covariance $\rho_{C1,C2}$, on $Y_1$ and $Y_2$. The effect sizes $\gamma_1, \gamma_2, \gamma_{C1}$ and $\gamma_{C2}$ were assumed to be normally distributed, and their variances were denoted as $\sigma_1^2, \sigma_2^2, \sigma_{C1}^2$ and $\sigma_{C2}^2$, respectively. Two reciprocal causal paths $(\delta_{12}$ and $\delta_{21})$ were specified between $Y_1$ and $Y_2$ (Fig. 1). Additionally, we considered systemic biases such as population stratification ($a_1$ and $a_2$) and sample overlap ($\rho_0$) in variance and covariance estimates for phenotype $Y_1$ and $Y_2$ in the model. From the above parameters, we also calculated the genetic correlation ($r_g$) between the two phenotypes.

From these assumptions, we derived the bivariate distribution of the marginal effect size estimates of the two phenotypes from their respective GWAS. The covariance matrix of this bivariate distribution is determined by the mixture distribution of the direct effect sizes $\gamma$, the reciprocal causal effects $\delta$ and the nuisance parameters $a_1$, $a_2$, and $\rho_0$ (see the "Methods" section). We then calculated the composite likelihood function for all available SNPs to estimate the two causal effects as well as other nuisance parameters by an expectation-maximization (EM) algorithm, together with robust sandwich estimates of their standard errors. Besides the above full model scenario, we further considered four sub-model scenarios in which one or two components were absent (Supplementary Fig. 1). Under these sub-model scenarios, we found that full model estimation often produced poor estimates, and developed a method based on model averaging to achieve more robust model fitting.

We then performed comprehensive simulations, including the full model and sub-model scenarios, to evaluate the estimation accuracy, Type I error rate, and statistical power of MRCI, compared to existing MR methodologies.

### Simulation results
**Estimation and hypothesis testing under the full model.** We compared the accuracy of the causal estimates between MRCI and IV-based MR methods under several simulated low polygenicity scenarios in which the true IVs could be considered nearly independent to satisfy the assumption of classic MR methods. MRCI produced nearly unbiased estimates for both causal directions under both independent and correlated pleiotropy scenarios. Not surprisingly, using exposure-specific true causal SNPs as IVs in IV-based MR methods always obtained unbiased estimates. However, many IV-based MR methods generated biased estimates when GWAS significant SNPs for the exposure phenotype were used as IVs, especially under scenarios with correlated pleiotropy, except for Weighted Mode. In these scenarios, excluding IVs which showed strong GWAS association with the outcome phenotype reduced the magnitude of the bias (Fig. 2a–c and Supplementary Fig. 2).

MRCI achieved correct control of the Type I error rates under all scenarios when the exposure phenotype had no effect on the outcome phenotype. Not surprisingly, IV-based MR methods also correctly controlled Type I error rates in the perfect scenario where all IVs were exposure-specific true causal SNPs. However, when IVs were selected according to statistical significance in GWAS of the exposure phenotype, most of the MR methods showed severely inflated Type I error rates in simulations with correlated pleiotropy. Excluding SNPs which showed a strong association with the outcome phenotype decreased the Type I error rates, sometimes to the correct level (Weighted Median), when correlated pleiotropy was present (Fig. 2d and Supplementary Tables 2–4).

In terms of statistical power under the alternative hypothesis, we only considered MR methods (MR-Egger, Weighted Mode, and MRMix) that demonstrated correct control of Type I error rates when using statistically significant exposure-associated SNPs as IVs. MRCI and MR methods achieved comparable power in low polygenicity simulations: as high as almost 1 in the stronger causal direction and slightly lower in the weaker causal direction. For IV-based MR methods, excluding potential outcome-associated significant SNPs did not guarantee a sufficient power increase in correlated pleiotropy scenarios (Supplementary Tables 3–7).

Additionally, simulations of other scenarios, including high polygenicity and asymmetry between the two phenotypes (different levels of heritability and polygenicity, and where one phenotype was binary and the other quantitative) showed that MRCI achieved nearly unbiased estimates, well-controlled Type I error rates, and favorable statistical power in both causal directions (Supplementary Table 8). Sample overlap was reflected in the estimates of the nuisance parameter $\rho_0$ and had no obvious effects on causal estimates in the simulation (Supplementary Fig. 3). Genetic correlation estimates calculated

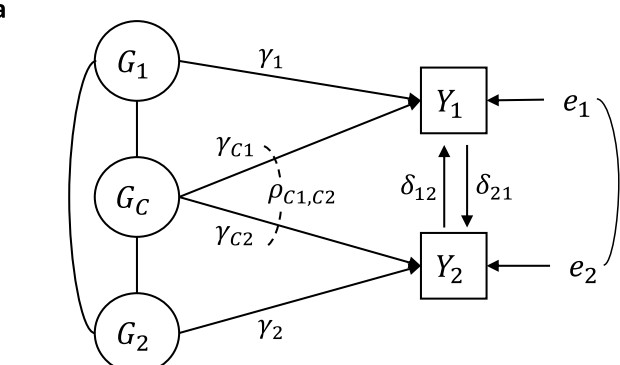

**a**

**b**

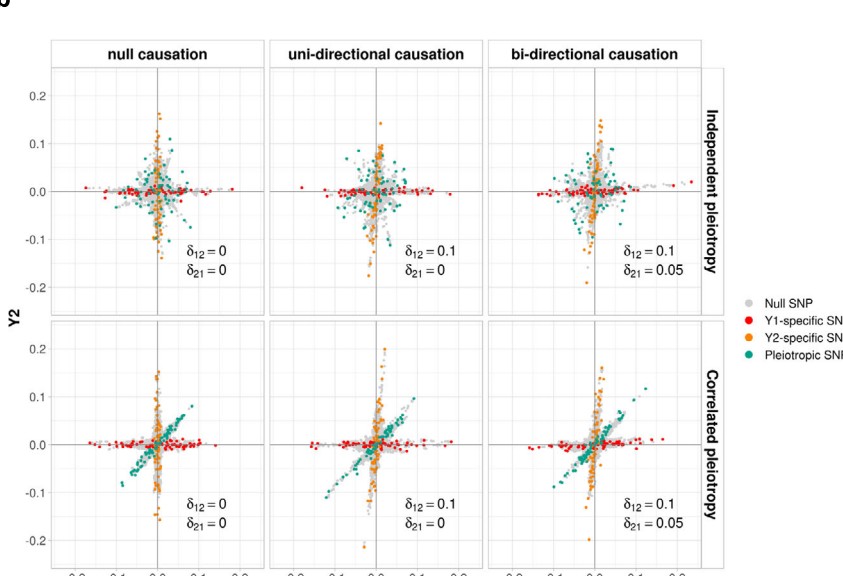

**Fig. 1 | Schematic diagram of MRCI. a** $Y_1$ and $Y_2$ represent a pair of phenotypes. Genotypes can be divided into four components: $Y_1$-specific causal SNPs ($G_1$), $Y_2$-specific causal SNPs ($G_2$), pleiotropic causal SNPs ($G_C$) and null SNPs ($G_0$, not shown). Lines connecting these genotypes represent the LD correlation between SNPs. Arrow lines from genotype to phenotype represent the direct effect of corresponding SNPs ($\gamma_1$, $\gamma_2$, $\gamma_{C1}$ and $\gamma_{C2}$) on the phenotypes. Covariance ($\rho_{C1,C2}$) between $\gamma_{C1}$ and $\gamma_{C2}$ is allowed. Arrow lines between the two phenotypes represent the reciprocal causal paths ($\delta_{12}$ and $\delta_{21}$). Non-additive genetic effects on phenotypes are represented by $e_1$ and $e_2$. In real situations, one or two components in the model could be absent and our method could handle these sub-model scenarios in a robust way. **b** Illustration of several representative simulation scenarios. simIDs are LoS1, LoS3, LoS7 (upper panel from left to right), LoS2, LoS4, LoS8 (lower panel from left to right) (see Supplementary Table 1). $x$-axis and $y$-axis show the standardized effect size estimates for GWAS $Y_1$ and $Y_2$, respectively. Red, orange, green, and gray points represent $Y_1$-specific, $Y_2$-specific, pleiotropic, and null SNPs in the simulation respectively. For null causation, $\delta_{12} = \delta_{21} = 0.0$; for uni-directional causation, $\delta_{12} = 0.1$, $\delta_{21} = 0.0$; for bi-directional causation, $\delta_{12} = 0.1$, $\delta_{21} = 0.05$. For independent pleiotropy, $\rho_{C1,C2} = 0.0$; for correlated pleiotropy, $\rho_{C1,C2} = 0.1$. In these plots, the mixing proportions for non-null components were set as $\pi_1 = \pi_2 = \pi_c = 1 \times 10^{-4}$, and the heritabilities contributed by $Y_1$-specific, $Y_2$-specific, and pleiotropic SNPs were set as 0.3, 0.3, and 0.1, respectively (see Supplementary Fig. 1 for other simulated scenarios).

from the MRCI parameter estimates were highly consistent with those obtained from LD Score regression (LDSC)[1] (Supplementary Table 9). Simulations with different sample sizes show that when the sample sizes of the GWAS of the two phenotypes are unequal, the accuracy of the causal effect estimate from the phenotype with the larger sample size to the phenotype with the smaller sample size is disproportionately reduced compared to the causal effect estimate in the other direction (Supplementary Table 10).

**Estimation and hypothesis testing under sub-models.** When the true model does not contain all four SNP components, there is a risk of incorrect inference from estimation that assumes the full model. To examine the robustness of MRCI, we performed estimations under four sub-models which allowed for the absence of one or two specific components. Writing the full model as $s_{1,2,C}$ (indicating the presence of

$Y_1$-specific SNPs, $Y_2$-specific SNPs, and pleiotropic SNPs, as well as the omnipresent null SNPs), the four sub-models were $s_{1,C}$, $s_{2,C}$, $s_{1,2}$ and $s_C$ (indicating the presence of the corresponding SNP components). We performed simulations to investigate model fitting performance under these sub-models, excluding $s_{1,C}$ as it is equivalent to $s_{2,C}$ (Supplementary Fig. 1 and Table 1). We found that estimation assuming the full model often did not correctly infer the absent component, which resulted in biased causal effect estimates and inflated Type I error rates (Supplementary Fig. 4a).

While estimation under the true model performed well in simulations, the true model is not known when applied to real data. Accordingly, to improve the robustness of MRCI we implemented an additional model fitting procedure which involved estimation under each sub-model and the full model, and performed model averaging of parameter estimates with weights that optimized the composite

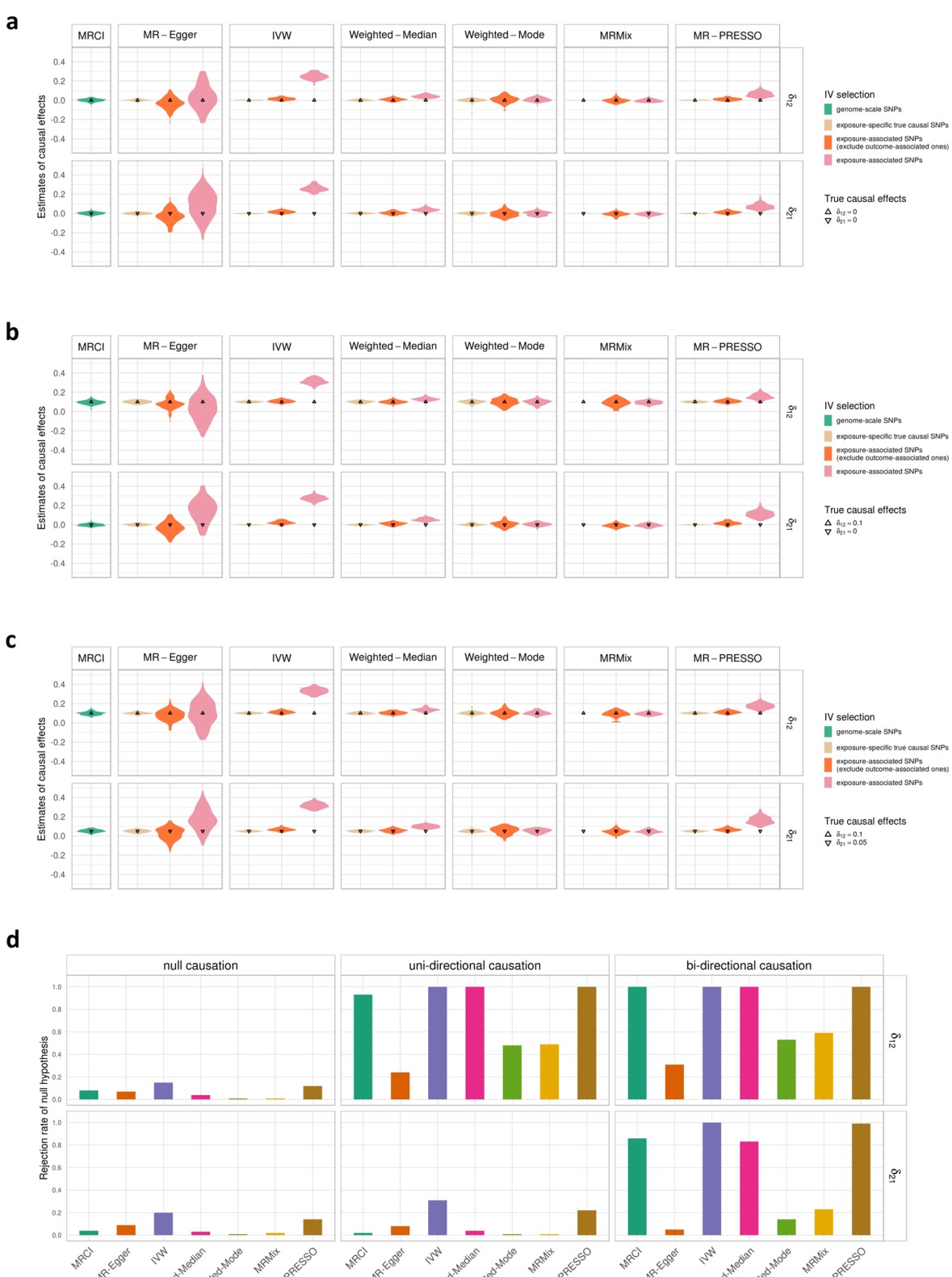

likelihood. Simulation results showed that this procedure usually assigned low weights to incorrect sub-models (Supplementary Fig. 4b). When the averaged model was preferred, which was usually the case when the exposure-specific component is absent (Supplementary Fig. 4c), parameter estimates from the model averaging procedure were adopted, instead of the full model estimates. In simulations under null, uni-directional, and bi-directional causation models, this method

gave nearly unbiased estimates of the reciprocal causal effects (Fig. 3a) and correct control of Type I error rates when a causal effect was absent, and acceptable power when a causal effect was present and the model included exposure-specific SNPs (Fig. 3b). We also tested sub-model scenarios with unbalanced pleiotropy effects and observed similar estimation results of the reciprocal causations (Supplementary Fig. 5).

**Fig. 2 | Comparison of estimates for the reciprocal causal effect by our method and instrumental variable (IV)-based MR methods in simulations with correlated pleiotropy. a** Estimates in null causation scenarios ($\delta_{12} = \delta_{21} = 0.0$, LoS2). **b** Estimates in uni-directional causation scenarios ($\delta_{12} = 0.1$ and $\delta_{21} = 0.0$, LoS4). **c** Estimates in bi-directional causation scenarios ($\delta_{12} = 0.1$ and $\delta_{21} = 0.05$, LoS8). **d** Rejection rates of the null hypothesis in bi-directional, uni-directional, and null scenarios. In plots **a**–**c**, our method took genome-scale SNPs for estimation and produced nearly unbiased estimates in different scenarios; the true values of $\delta_{12}$ and $\delta_{21}$ are indicated by up- and down-pointing triangles, respectively. For MR methods, IVs were selected in three ways: (1) use the exposure-specific true causal SNPs in the simulation as IVs; (2) use exposure-associated SNPs ($p$-value $< 5 \times 10^{-8}$)

after clumping but exclude potential outcome-associated SNPs (defined as $p$-value $< 5 \times 10^{-5}$ with the outcome); (3) use significant exposure-associated SNPs after clumping regardless of their association with outcome. In plot **d**, the exclusion criteria were applied to IV-based MR methods and our method shows well-controlled Type I error rates and adequate power. In these plots, $\rho_{C1,C2}$ for the correlated pleiotropy is 0.1; the mixing proportions for non-null components were $\pi_1 = \pi_2 = \pi_c = 1 \times 10^{-4}$; the heritabilities contributed by $Y_1$-specific, $Y_2$-specific, and pleiotropic SNPs were 0.3, 0.3, and 0.1, respectively (note: The selection of exposure-specific true causal SNPs was not applied to MRMix due to its assumption of a normal-mixture distribution).

## Table 1 | Descriptive summary statistics of included traits

| Traits | Total reported sample size | Number of reported cases | Number of reported controls | Number of independent significant SNPs/loci | Variance explained by significant SNPs/loci (%) | Publication |
|---|---|---|---|---|---|---|
| CAD | 148,172 | 10,801 | 137,371 | 66 | 21.2 | Nelson et al. (2017)[33] |
| IS | 440,328 | 34,217 | 406,111 | 2 | 0.6–1.8 | Malik et al. (2018)[34] |
| T2D | 898,130 | 74,124 | 824,006 | 243 | 18.0[a] | Mahajan et al. (2018)[35] |
| BirthWeight | 298,142 | – | – | 190 | 7.0 | Warrington et al. (2019)[36] |
| BMI | 681,275 | – | – | 941 | 6.0 | Yengo et al. (2018)[37] |
| BodyFat | 89,297 | – | – | 7 | 0.6 | Lu et al. (2016)[38] |
| Height | 693,529 | – | – | 3290 | 24.6 | Yengo et al. (2018)[37] |
| FastGluc | 140,595[b] | – | – | 7[c] | – | Lagou et al. (2021)[39] |
| FastInsulin | 98,210[b] | – | – | 1[c] | – | Lagou et al. (2021)[39] |
| HDL | 188,577 | – | – | 70 | 1.6 | Willer et al. (2013)[40] |
| LDL | 188,577 | – | – | 57 | 2.4 | Willer et al. (2013)[40] |
| Triglycerides | 188,577 | – | – | 37 | 2.1 | Willer et al. (2013)[40] |
| CRP | 418,642 | – | – | 526 | 13.0 | Han et al. (2020)[41] |
| CigPerDay | 337,334 | – | – | 55 | ~1.1 | Liu et al. (2019)[42] |
| DrinksPerWeek | 941,280 | – | – | 99 | ~0.2 | Liu et al. (2019)[42] |
| pulsePressure | 757,601 | – | – | 143/62[d] | 2.6 | Evangelou et al. (2018)[43] |
| dBP | 757,601 | – | – | 130/120[d] | 4.5 | Evangelou et al. (2018)[43] |
| sBP | 757,601 | – | – | 183/81[d] | 4.8 | Evangelou et al. (2018)[43] |
| MDD | 500,199 | 170,756 | 329,443 | 102[e] | 1.5–3.2% | Howard et al. (2019)[44] |

*CAD* coronary artery disease, *IS* ischemic stroke, *T2D* type 2 diabetes, *BirthWeight* birth weight, *BMI* body mass index, *BodyFat* body fat percentage, *FastGluc* fasting glucose level, *FastInsulin* fasting insulin level, *HDL* high-density lipoprotein cholesterol level, *LDL* low-density lipoprotein cholesterol level, *Triglycerides* triglyceride level, *CRP* C-reactive protein level, *CigPerDay* cigarettes per day, *DrinksPerWeek* drinks per week, *pulsePressure* pulse pressure, *dBP* diastolic blood pressure, *sBP* systolic blood pressure, *MDD* major depressive disorder.
[a]Trait variance explained by all GWAS SNPs/loci.
[b]Maximum number of samples used for meta-analysis, not only European population.
[c]Discoveries in this study.
[d]Two-stage/one-stage. One-stage analysis criteria: $p$-value $< 5 \times 10^{-9}$ for the discovery meta-analysis (757,601 samples), $p$-value $< 0.01$ for UKB (458,577 samples), $p$-value $< 0.01$ for ICBP (299,024 samples) and concordant direction of effect between UKB and ICBP. Two-stage analysis criteria: genome-wide significance in the combined meta-analysis, $p$-value $< 0.01$ in the replication meta-analysis and concordant direction of effect.
[e]For all meta-analysis, not specific to European populations.

**Comparison with CAUSE and MRMix.** CAUSE[13] and MRMix[12] are two recent methods that share similarities with MRCI. With large samples (50,000 for both phenotypes), when the true model was $s_{1,2,C}$ or $s_{1,2}$, simulation results showed that these two methods produced unbiased estimates and correct Type I error rates. However, under scenarios where trait-specific SNPs were absent for one or both phenotypes (i.e., $s_{2,C}$ and $s_C$) these methods obtained biased causal estimates and inflated Type I error rates for the causal direction in which the exposure-specific SNPs were absent (Fig. 4a, b). With small samples (20,000 for both phenotypes), CAUSE produced biased estimates and

increased Type I error rates, even under the full model. Under these small sample scenarios, both MRCI and MRMix maintained unbiased estimates and corrected Type I error rates, with MRCI achieving slightly smaller estimation variance and greater statistical power (Fig. 4c).

**Causation of risk factors on common diseases.** We applied MRCI to study causal relationships between 3 common diseases (coronary artery disease, CAD; ischemic stroke, IS; type 2 diabetes, T2D) and 16 putative risk factor phenotypes (Table 1). Figure 5 shows the estimated

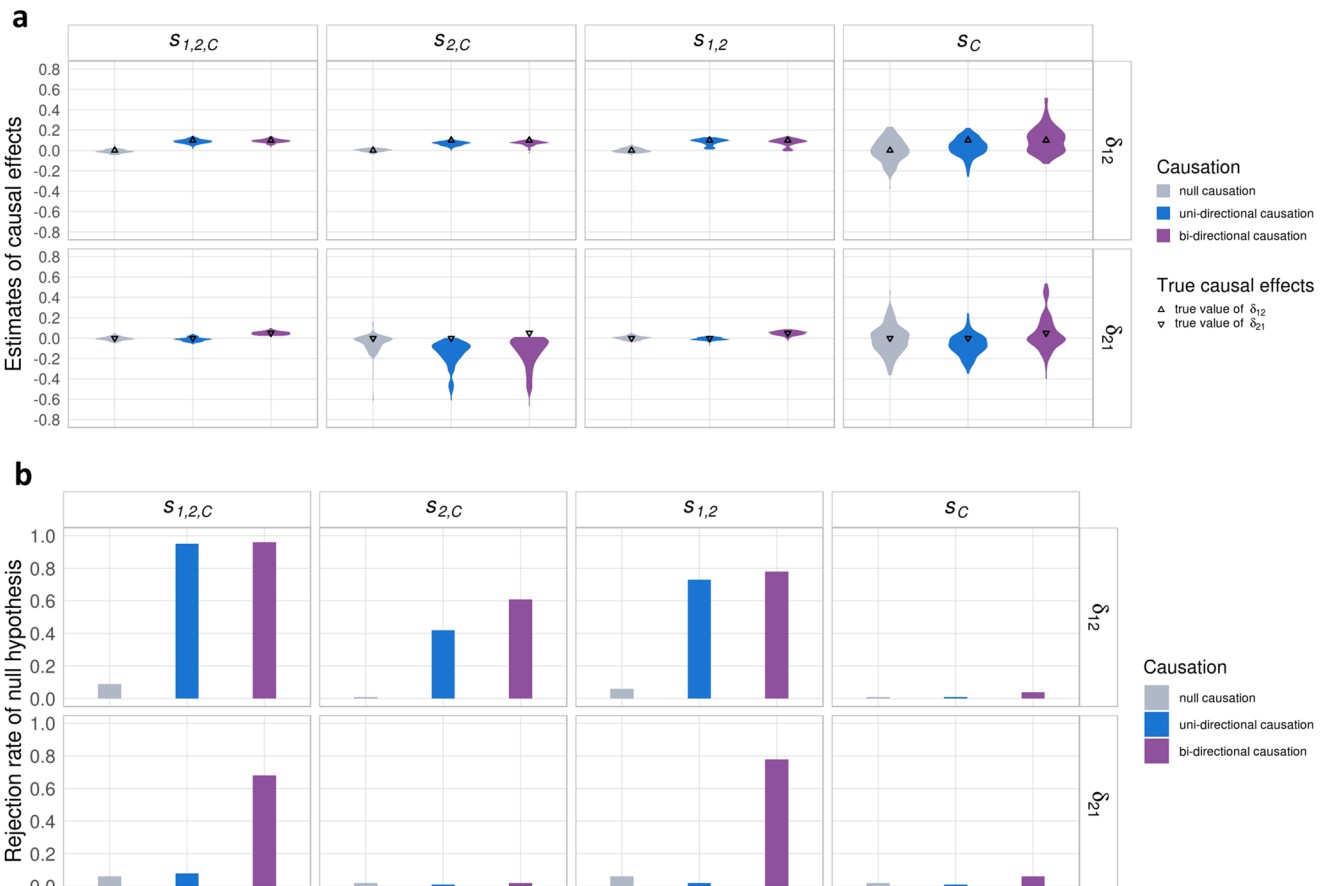

**Fig. 3 | Estimation after model averaging. a** Final estimates of causal effects under null, uni-directional, and bi-directional causations in four scenarios ($s_{1,2,C}$, $s_{2,C}$, $s_{1,2}$, and $s_C$). When the exposure-specific SNPs exist, the final estimates after model averaging still produced nearly unbiased estimates in simulated scenarios. For null causation (gray) $\delta_{12} = \delta_{21} = 0.0$; for uni-directional causation (blue), $\delta_{12} = 0.1$ and $\delta_{21} = 0.0$; for bi-directional causation (purple) $\delta_{12} = 0.1$ and $\delta_{21} = 0.05$. The true causation values of $\delta_{12}$ and $\delta_{21}$ are indicated by up- and down-pointing triangles, respectively. **b** Rejection rates of the null hypothesis of the final $\delta_{12}$ and $\delta_{21}$ estimates after model averaging in different scenarios. For the zero-effect causal direction, the Type I error rates were well-controlled; for the non-zero-effect causal direction, the presence of the exposure-specific SNPs showed reasonably good power, and the absence of the exposure-specific SNPs showed conservative power of estimation. In the simulations, the mixing proportions of the present component were $1 \times 10^{-3}$; the pleiotropic effects were correlated ($\rho_{C1,C2} = 0.1$); and the heritabilities contributed by $Y_1$-specific, $Y_2$-specific and pleiotropic SNPs (if present in the sub-model scenario) were 0.3, 0.3, and 0.1, respectively.

reciprocal causal effects ($\delta$) between these diseases and phenotypes. Considering that 96 tests were performed, the Bonferroni adjusted significance level is $5.0 \times 10^{-4}$. Causal influences significant at this level were from low-density lipoprotein level (LDL) to CAD ($\delta = 0.32$, 95% CI = [0.18,0.47], $p$-value = $1.7 \times 10^{-5}$), and from body mass index (BMI) to T2D ($\delta = 0.72$, 95% CI = [0.59,0.85], $p$-value = $3.8 \times 10^{-30}$). In the reverse direction, significant causal influences were detected from T2D to BMI ($\delta = -0.17$, 95% CI = [−0.24,−0.10], $p$-value = $2.2 \times 10^{-6}$) and fasting glucose ($\delta = 0.32$, 95% CI = [0.20, 0.44], $p$-value = $6.9 \times 10^{-8}$).

A number of nominally significant ($p$-value < 0.05) causal effects that did not survive multiple testing adjustments were detected (Fig. 5). These included effects of fasting glucose on CAD and LDL on both IS and T2D. Nominally significant effects in the reverse direction included CAD on BMI, LDL and diastolic blood pressure (dBP), and T2D on birth weight.

## Discussion

We have developed a new method, MRCI, to jointly estimate the reciprocal causal effects between two phenotypes using GWAS summary statistics and reference LD data. Our extensive simulation studies, under both independent pleiotropy and correlated pleiotropy scenarios, indicate that MRCI obtains nearly unbiased estimates of causation in both directions, and maintains well-controlled Type I error rates under the null hypothesis, even when most causal SNPs for

one or both phenotypes are pleiotropic. MRCI also achieves comparable statistical power to other methods when Type I error rates are controlled.

Our simulations showed that existing MR methods that require the selection of IVs produced unbiased estimates and correct Type I error rates when using genome-wide significant exposure-associated IVs, for scenarios with independent pleiotropy. However, some methods often produced biased estimates and severely inflated Type I error rates in the presence of correlated pleiotropy. For MR-Egger[8] and MR-PRESSO[9], designed to cope with horizontal pleiotropy under the InSIDE assumption, excluding potential outcome-associated SNPs according to a $p$-value cut-off helped control Type I error, but in practice, the optimal choice for cut-off is unknown and may vary depending on the genetic architecture of the phenotypes as well as GWAS sample sizes. These are important limitations as pleiotropy is common for complex traits[9], and independent pleiotropy is implausible[8].

The robustness of MRCI is partly derived from the explicit consideration of sub-models with absent SNP components during model fitting. When an SNP component is absent, estimation under the full model sometimes incorrectly infers the absent SNP components which can lead to spurious inferences on the causal effects. Thus, we perform estimation under both the full model and sub-models, but the goal is not to select a single best model. Rather, our aim is to obtain robust causal estimates. Based on simulation performance, our method of

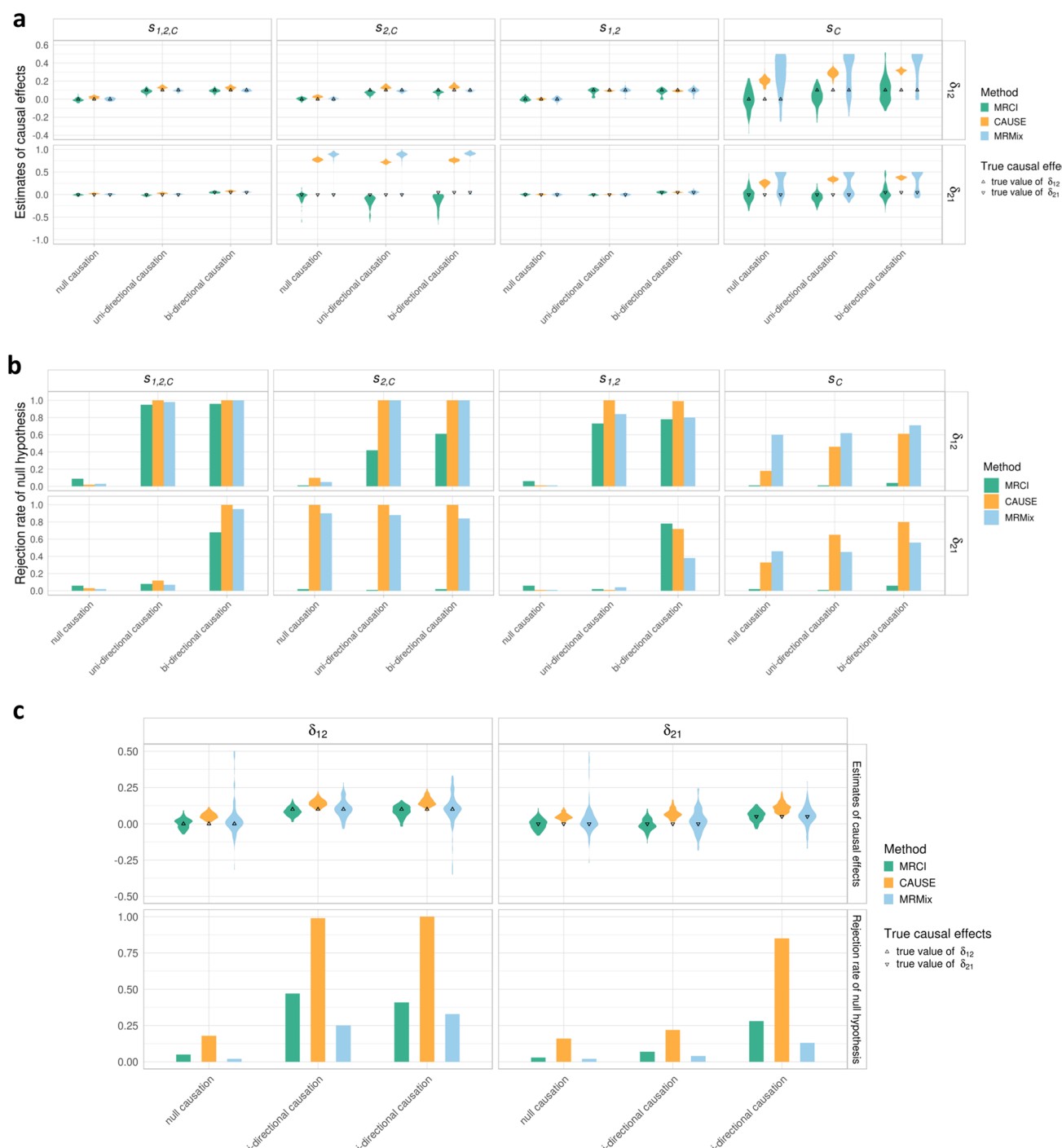

**Fig. 4 | Estimation comparison with CAUSE and MRMix. a** Estimates from MRCI (green), CAUSE (orange), and MRMix (blue) under null, uni-directional, and bi-directional causations in four scenarios ($s_{1,2,C}$, $s_{2,C}$, $s_{1,2}$, and $s_C$). CAUSE and MRMix produced severely over-estimated causal effects when exposure-specific SNPs were absent in the sub-model $s_{2,C}$ and $s_C$. For null causation, $\delta_{12} = \delta_{21} = 0.0$; for uni-directional causation, $\delta_{12} = 0.1$ and $\delta_{21} = 0.0$; for bi-directional causation, $\delta_{12} = 0.1$ and $\delta_{21} = 0.05$. The true causation values of $\delta_{12}$ and $\delta_{21}$ are indicated by up- and down-pointing triangles, respectively. **b** Rejection rates of the null hypothesis for $\delta_{12}$ and $\delta_{21}$ estimates from MRCI (green), CAUSE (orange), and MRMix (blue) in different simulated scenarios. CAUSE and MRMix produced inflated Type I error rates for the causal direction where exposure-specific SNPs were absent ($s_{2,C}$ and $s_C$). **c** Estimates of causal effects and rejection rates of null hypothesis from MRCI

(green), CAUSE (orange) and MRMix (blue) under null, uni-directional, and bi-directional causations in $s_{1,2,C}$ scenarios with small sample sizes (20,000 individuals). Decreased GWAS power led to over-estimates for CAUSE and larger estimation variance for MRMix. MRCI produced nearly unbiased estimates and correct Type I error rates. In these results, the estimates of MRCI came from the final estimates after model averaging; $p$-value thresholds for CAUSE and MRMix were $1 \times 10^{-3}$ and $5 \times 10^{-8}$, respectively. In the simulations, the mixing proportions of the present component were $1 \times 10^{-3}$; the pleiotropic effects were correlated ($\rho_{C1,C2} = 0.1$); the heritabilities contributed by $Y_1$-specific, $Y_2$-specific and pleiotropic SNPs (if present in the sub-model scenario) were 0.3, 0.3, and 0.1, respectively.

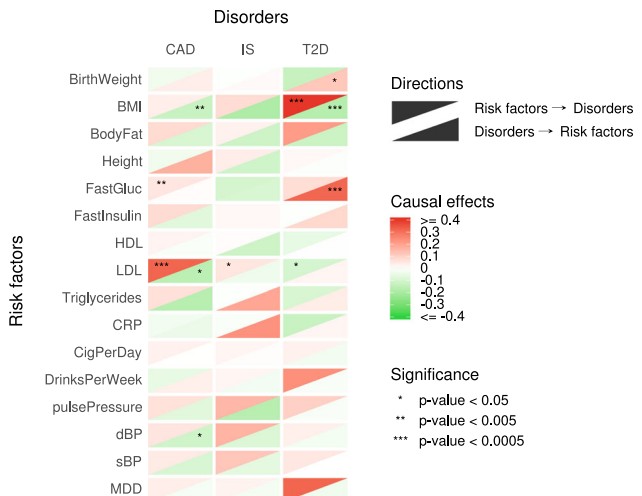

**Fig. 5 | Summary of estimates for three common diseases and sixteen risk factors.** In the figure, the upper triangle in a box represents the causal direction from the risk factor to the disorder, and the lower triangle represents the reverse causal direction. Darker color represents stronger causation estimates. Stars in the triangle represent different significance levels of the estimation. CAD coronary artery disease, IS any ischemic stroke, T2D type 2 diabetes, BirthWeight birth weight, BMI body mass index, BodyFat body fat percentage, FastGluc fasting glucose level, FastInsulin fasting insulin level, HDL high-density lipoprotein cholesterol level, LDL low-density lipoprotein cholesterol level, Triglycerides triglyceride level, CRP C-reactive protein level, CigPerDay cigarettes per day, DrinksPerWeek drinks per week, pulsePressure pulse pressure, dBP diastolic blood pressure, sBP systolic blood pressure, MDD major depressive disorder.

model averaging reduces the chance of such spurious inferences and produces unbiased estimates and correct control of Type I error rates even in these challenging scenarios. Two recent methods, CAUSE[13] and MRMix[12], which share some similarities with MRCI, nevertheless produced biased estimates and inflated Type I error rates when a trait-specific SNP component was absent.

MRCI also provides estimates for nuisance parameters such as heritabilities and genetic correlation, which may help clarify the genetic architecture of the two phenotypes. In our model, the parameters for the population stratification are also considered nuisance parameters, which behave similarly to genomic control when testing in mixed population simulations and barely affect the causal estimates in the simulated scenarios (Supplementary Fig. 6). Since most reported GWAS would have minimized population stratification effects prior to association testing, including these parameters during estimation represents an additional safeguard. The genetic correlation estimates from MRCI are consistent with results obtained from LDSC. Moreover, our model takes the sample overlap into consideration, including complete overlap which enables GWAS results obtained from the same sample (e.g., UK Biobank) to be analyzed for causal relationships.

When applied to real GWAS data, MRCI was able to identify several well-established causal relationships at a Bonferroni-corrected significance level, including a reciprocal causal relationship between body mass index and type 2 diabetes. Being overweight is known to be strongly associated with an increased risk of type 2 diabetes[17–19], while weight loss is a known clinical feature of type 2 diabetes[20]. MRCI also confirmed low-density lipoprotein level but not high-density lipoprotein level as a causal factor for coronary artery disease[21–23]. MRCI did not detect additional causal effects at the Bonferroni-corrected significance level, which may reflect the ability of MRCI to control the Type I error rate. It is also possible that GWAS for some of the phenotypes may not have adequate statistical power because of small sample sizes or high polygenicity.

The current formulation of MRCI has certain limitations. Our assumption of bivariate normality of standardized marginal effects in the four SNP components may not hold in practice, and serious violation of this assumption may affect the performance of the method. Further simulations to evaluate the sensitivity of the method to violations of this assumption are desirable. A second limitation is that we used composite likelihood for model fitting because marginal SNP effects are correlated. This necessitated the use of robust sandwich standard errors, which may lead to conservative, less powerful tests. Finally, in the scenario where pleiotropic effects are very strong and nearly no causal SNPs contribute specifically to the exposure, MRCI will typically produce very large standard errors for the causal path estimate from exposure to the outcome, so there is little power to detect such a causal effect even when it is present. However, this situation of having almost no SNPs that can be used as valid IVs could be a limitation for the MR approach in general, rather than for MRCI specifically.

## Methods

### Data and the full model

MRCI uses the GWAS summary statistics of the two phenotypes ($Y_1$ and $Y_2$) and reference LD information (~1 million SNPs from 1000 Genomes Project data). For a pair of reciprocal causal phenotypes (Fig. 1a), each available SNP belongs to one of four mutually exclusive SNP components:

- $Y_1$-specific component ($G_1$): SNPs that contribute directly to $Y_1$ only;
- $Y_2$-specific component ($G_2$): SNPs that contribute directly to $Y_2$ only;
- pleiotropic component ($G_C$): SNPs that contribute directly to both phenotypes;
- null component ($G_0$): SNPs with no direct effects on either phenotype.

The proportions of all SNPs in the four components are $\pi_1, \pi_2, \pi_c$ and $\pi_0$. The values of the two phenotypes in an individual are given by:

$$Y_1 = \delta_{12} Y_2 + \sum_{i \in G_1} \gamma_{1i} X_i + \sum_{l \in G_c} \gamma_{C1l} X_l + e_1$$

$$Y_2 = \delta_{21} Y_1 + \sum_{j \in G_2} \gamma_{2j} X_j + \sum_{l \in G_c} \gamma_{C2l} X_l + e_2$$

Here, $Y_1$ and $Y_2$ are the two standardized phenotypes; $\delta_{12}$ is the causal effect $Y_2 \to Y_1$ and $\delta_{21}$ is the causal effect $Y_1 \to Y_2$; $X_i, X_j$ and $X_l$ represent the standardized genotype of $i$th, $j$th and $l$th SNP in $G_1, G_2$ and $G_c$ component, respectively; $\gamma_{1i}, i \in G_1$ and $\gamma_{2j}, j \in G_2$ denote the direct effect sizes of phenotype-specific SNPs for $Y_1$ and $Y_2$; $\gamma_{C1l}$ and $\gamma_{C2l}, l \in G_c$ denote the direct effect sizes of pleiotropic SNPs for $Y_1$ and $Y_2$, respectively; $e_1$ and $e_2$ are residual effects. Since the SNPs in $G_0$ have no direct effects on either phenotype, they are not included. For the direct effect sizes of SNPs, we assume $\gamma_{1i} \sim N(0, \sigma_1^2)$, $\gamma_{2j} \sim N(0, \sigma_2^2)$ and $\begin{pmatrix} \gamma_{C1l} \\ \gamma_{C2l} \end{pmatrix} \sim N\left[ \begin{pmatrix} 0 \\ 0 \end{pmatrix}, \begin{pmatrix} \sigma_{C1}^2 & \rho_{C1,C2} \\ \rho_{C1,C2} & \sigma_{C2}^2 \end{pmatrix} \right]$ for $i \in G_1, j \in G_2$ and $l \in G_c$, where $\sigma_1^2$ and $\sigma_2^2$ denote the per-SNP variances of $G_1$ and $G_2$, and $\sigma_{C1}^2$ and $\sigma_{C2}^2$ the per-SNP variances of $G_C$ for $Y_1$ and $Y_2$ respectively, with covariance $\rho_{C1,C2}$. Therefore, the model encompasses both independent pleiotropy ($\rho_{C1,C2} = 0$) and correlated pleiotropy ($\rho_{C1,C2} \neq 0$).

The above formulae can be expressed in the bivariate matrix form $\boldsymbol{Y} = [\boldsymbol{I} - \boldsymbol{\Delta}]^{-1} \sum_h \sum_{kh} \boldsymbol{\Gamma}_k^{(h)} X_k + \boldsymbol{\varepsilon}$, where $\boldsymbol{Y}$ is a $2 \times 1$ vector of the two standardized phenotypes, $\boldsymbol{I}$ is the $2 \times 2$ identity matrix, $\boldsymbol{\Delta} = \begin{pmatrix} 0 & \delta_{12} \\ \delta_{21} & 0 \end{pmatrix}$,

$\boldsymbol{\Gamma}_k^{(h)}$ is the direct effect of the $k$th SNP on the phenotypes depending on its component membership $h$, i.e. $h \in (G_0, G_1, G_2, G_C)$ and $\boldsymbol{\Gamma}_k^{(G_0)} = \begin{pmatrix} 0 \\ 0 \end{pmatrix}$, $\boldsymbol{\Gamma}_k^{(G_1)} = \begin{pmatrix} \gamma_{1k} \\ 0 \end{pmatrix}$, $\boldsymbol{\Gamma}_k^{(G_2)} = \begin{pmatrix} 0 \\ \gamma_{2k} \end{pmatrix}$, $\boldsymbol{\Gamma}_k^{(G_C)} = \begin{pmatrix} \gamma_{C1k} \\ \gamma_{C2k} \end{pmatrix}$, $X_k$ is the standardized genotype for the $k$th SNP and $\boldsymbol{\varepsilon}$ is the residual effects of any non-additive genetic and environmental factors. This expression is similar to that used for reciprocal causal modeling in twin data[24] and requires that $|\delta_{12}\delta_{21}| < 1$ to ensure that the reciprocal causation between the two phenotypes results in a steady state rather than unlimited growth[25]. In practice, we constrain both $\delta_{12}$ and $\delta_{21}$ to be between −1.0 and 1.0, since both phenotypes are standardized variances as 1.0, ensuring the inequality.

If we define $\boldsymbol{\beta}_k^{(h)} = [\boldsymbol{I} - \boldsymbol{\Delta}]^{-1}\boldsymbol{\Gamma}_k^{(h)}$ as a $2 \times 1$ vector of the component-dependent joint effect sizes for the $k$th SNP, including any effect mediated through one phenotype on the other, then the model can be expressed as $\boldsymbol{Y} = \sum \boldsymbol{\beta}_k^{(h)} X_k + \boldsymbol{\varepsilon}$. For each phenotype, the marginal effect of SNP $k$ is $\tau_k = \sum_{i=1}^{N_k^*} \beta_i \rho_{ki}$, where $N_k^*$ is the total number of SNPs tagged by the $k$th SNP, $\rho_{ki}$ is the LD correlation between $k$th and $i$th SNP, and $\beta_i$ is the joint effect size of the $i$th SNP tagged by the $k$th SNP[26], taking account of the reciprocal causations.

## Composite likelihood estimator

We assume that the estimates of the marginal effects of the $k$th SNP from the GWAS summary statistics have a bivariate normal distribution:

$$\hat{\boldsymbol{\tau}}_k = \begin{pmatrix} \hat{\tau}_{1k} \\ \hat{\tau}_{2k} \end{pmatrix} \sim \sum_{\mathbb{N}_k} \Pr(\mathbb{N}_k) N\left[ \begin{pmatrix} 0 \\ 0 \end{pmatrix}, \begin{pmatrix} \sigma_{\hat{\tau}_{1k}}^2 & \rho_{\hat{\tau}_{1k},\hat{\tau}_{2k}} \\ \rho_{\hat{\tau}_{1k},\hat{\tau}_{2k}} & \sigma_{\hat{\tau}_{2k}}^2 \end{pmatrix} \right]$$

In the formula, $\hat{\tau}_{1k}$ and $\hat{\tau}_{2k}$ represent the marginal effect size estimates of the $k$th SNP from GWAS summary statistics of phenotype $Y_1$ and $Y_2$, respectively. $\mathbb{N}_k = \left( N_k^{(G_1)}, N_k^{(G_2)}, N_k^{(G_C)}, N_k^{(G_0)} \right)$ are the possible combinations of counts of SNPs in LD with SNP $k$ that belong to the four SNP components. The probability of each combination, $\Pr(\mathbb{N}_k)$, can be calculated based on the multinomial distribution with total counts $N_k^* = \sum_h N_k^{(h)}$ over $h \in (G_0, G_1, G_2, G_C)$, where $N_k^{(h)}$ is a latent variable denoting the number of $h$-component SNPs tagged by the $k$th SNP. $\sigma_{\hat{\tau}_{1k}}^2$, $\sigma_{\hat{\tau}_{2k}}^2$ and $\rho_{\hat{\tau}_{1k},\hat{\tau}_{2k}}$ can be derived from the relationships between the marginal and joint regression coefficients in mixture form (see Supplementary Note):

$$\sigma_{\hat{\tau}_{1k}}^2 \approx \frac{\sigma_1^2}{(1-\delta_{12}\delta_{21})^2}\frac{N_k^{(G_1)}}{N_k^*}\ell_k + \frac{\delta_{12}^2\sigma_2^2}{(1-\delta_{12}\delta_{21})^2}\frac{N_k^{(G_2)}}{N_k^*}\ell_k$$
$$+ \frac{\left[\sigma_{C1}^2 + \delta_{12}^2\sigma_{C2}^2 + 2\delta_{12}\rho_{C1,C2}\right]}{(1-\delta_{12}\delta_{21})^2}\frac{N_k^{(G_C)}}{N_k^*}\ell_k + a_1 + 1/n_1$$

$$\sigma_{\hat{\tau}_{2k}}^2 \approx \frac{\delta_{21}^2\sigma_1^2}{(1-\delta_{12}\delta_{21})^2}\frac{N_k^{(G_1)}}{N_k^*}\ell_k + \frac{\sigma_2^2}{(1-\delta_{12}\delta_{21})^2}\frac{N_k^{(G_2)}}{N_k^*}\ell_k$$
$$+ \frac{\left[\sigma_{C2}^2 + \delta_{21}^2\sigma_{C1}^2 + 2\delta_{21}\rho_{C1,C2}\right]}{(1-\delta_{12}\delta_{21})^2}\frac{N_k^{(G_C)}}{N_k^*}\ell_k + a_2 + 1/n_2$$

$$\rho_{\hat{\tau}_{1k},\hat{\tau}_{2k}} \approx \frac{\delta_{21}\sigma_1^2}{(1-\delta_{12}\delta_{21})^2}\frac{N_k^{(G_1)}}{N_k^*}\ell_k + \frac{\delta_{12}\sigma_2^2}{(1-\delta_{12}\delta_{21})^2}\frac{N_k^{(G_2)}}{N_k^*}\ell_k$$
$$+ \frac{\left[\delta_{21}\sigma_{C1}^2 + \delta_{12}\sigma_{C2}^2 + (1+\delta_{12}\delta_{21})\rho_{C1,C2}\right]}{(1-\delta_{12}\delta_{21})^2}\frac{N_k^{(G_C)}}{N_k^*}\ell_k + \rho_0$$

where $\ell_k$ is the LD score for the $k$th SNP; $a_1$ and $a_2$ are additional inflation factors accounting for systematic bias in variance estimates (e.g., due to population stratification) for phenotype $Y_1$ and $Y_2$ respectively[26]; $\rho_0$ is a factor accounting for bias in the covariance estimates (e.g., due to sample overlap); $n_1$ and $n_2$ are the sample sizes for the two GWAS of the two phenotypes. In this way, the partitioned genetic variances, LD conditions as well as reciprocal causal effects are reflected in the assumed bivariate distribution.

The likelihood for the summary-statistic of the $k$th SNP is then: $L(\boldsymbol{\theta};\hat{\boldsymbol{\tau}}_k) = p(\hat{\boldsymbol{\tau}}_k, |, \boldsymbol{\theta}) = \sum_{\mathbb{N}_k} \Pr(\mathbb{N}_k) f(\hat{\tau}_{1k}, \hat{\tau}_{2k})$, where $f(\hat{\tau}_{1k}, \hat{\tau}_{2k})$ is the density function of a bivariate normal distribution with parameters $\boldsymbol{\theta} = (\pi_1, \pi_2, \pi_c, \sigma_1^2, \sigma_2^2, \sigma_{C1}^2, \sigma_{C2}^2, \rho_{C1,C2}, \delta_{12}, \delta_{21}, a_1, a_2, \rho_0)$. Thus, the composite log-likelihood function is in the form:

$$CL(\boldsymbol{\theta};\hat{\boldsymbol{\tau}}) = \sum_{k=1}^K \log L(\boldsymbol{\theta};\hat{\boldsymbol{\tau}}_k) = \sum_{k=1}^K \log\left[ \sum_{\mathbb{N}_k} \Pr(\mathbb{N}_k) f(\hat{\tau}_{1k}, \hat{\tau}_{2k}) \right]$$

Thus, the maximum composite likelihood estimator is given by

$$\hat{\boldsymbol{\theta}} = \underset{\boldsymbol{\theta}}{argmax}\, CL(\boldsymbol{\theta};\hat{\boldsymbol{\tau}})$$

## Parameter estimation and testing

The reciprocal causal paths ($\delta_{12}$ and $\delta_{21}$), together with nuisance parameters, are estimated by maximizing the likelihood with an expectation-maximization (EM) algorithm. For each M-step in the EM, parameters for mixing proportions ($\pi_1, \pi_2$ and $\pi_c$) are estimated according to the closed form, while the remaining parameters (i.e. $\sigma_1^2, \sigma_2^2, \sigma_{C1}^2, \sigma_{C2}^2$, $\rho_{C1,C2}, \delta_{21}, \delta_{12}, a_1, a_2$ and $\rho_0$) are estimated by Nelder–Mead optimization (see Supplementary Note).

The standard errors of each parameter are calculated using a sandwich variance estimator adapted from Zhang et al.[26] since sandwich estimators can provide valid variance estimation and tolerate the possible misspecification of the model. Briefly, if we define $l_k(\boldsymbol{\theta}) = \log L(\boldsymbol{\theta};\hat{\boldsymbol{\tau}}_k)$ then the score function could be written as in the form: $U(\boldsymbol{\theta}) = \frac{\partial CL(\boldsymbol{\theta};\hat{\boldsymbol{\tau}}_k)}{\partial\boldsymbol{\theta}} = \sum_{k=1}^K \frac{\partial \log L(\boldsymbol{\theta};\hat{\boldsymbol{\tau}}_k)}{\partial\boldsymbol{\theta}} = \sum_{k=1}^K \frac{\partial l_k(\boldsymbol{\theta})}{\partial\boldsymbol{\theta}} = \sum_{k=1}^K U_k(\boldsymbol{\theta})$, where $U_k(\boldsymbol{\theta})$ is the score vector for the $k$th SNP. Then the sandwich form of variance-covariance matrix for the estimate $\hat{\boldsymbol{\theta}}$ is $var(\hat{\boldsymbol{\theta}}) = I^{-1}(\boldsymbol{\theta})J(\boldsymbol{\theta})I^{-1}(\boldsymbol{\theta})$, which can be estimated by plugging in the estimated parameter values $\hat{\boldsymbol{\theta}}$ in lieu of $\boldsymbol{\theta}$. Thus, we can get $\hat{I}(\hat{\boldsymbol{\theta}}) = -\sum_{k=1}^K \frac{\partial l_k^2(\boldsymbol{\theta})}{\partial\boldsymbol{\theta}\partial\boldsymbol{\theta}^T}|_{\boldsymbol{\theta}}$ and $\hat{J}(\hat{\boldsymbol{\theta}}) = \sum_{k=1}^K U_k(\hat{\boldsymbol{\theta}})\bar{U}_k^T(\hat{\boldsymbol{\theta}})$, where $\bar{U}_k^T(\hat{\boldsymbol{\theta}}) = \sum_{k'\in\mathbb{N}_k} U_{k'}(\hat{\boldsymbol{\theta}})$ is the sum of likelihood scores for all SNPs tagged by the $k$th SNP. In practice, we used the symmetric derivative to obtain the derivatives for parameter estimates (see Supplementary Note). To determine the significance of estimates for parameters of interest, estimates and the corresponding standard errors are converted to $\chi^2$ statistics.

## Sub-models and model averaging

To increase the robustness of MRCI in real situations, we performed model fitting not only to the full model (referred to as the $s_{1,2,C}$ model) but also to four sub-models, each of which has one or two absent SNP

components: (I) the $s_{2,C}$ model excludes $G_1$ SNPs; (II) the $s_{1,C}$ model excludes $G_2$ SNPs; (III) the $s_C$ model excludes both $G_1$ and $G_2$ SNPs; and (IV) the $s_{1,2}$ model excludes $G_C$ SNPs. For each sub-model, the parameters are estimated using the same composite likelihood estimator as above, only that the parameters of the corresponding absent components are set as zero during the EM process and are thus excluded in variance calculation. Accordingly, we define the complete model set $S = (s_{1,2,C}, s_{2,C}, s_{1,C}, s_{1,2}, s_C)$.

After estimating all five models, we calculated weighted averages of the parameter estimates of these models to obtain an "averaged" model, and calculated its composite likelihood. The averaged estimate for the $j$th parameter can be written as $\hat{\theta}_{j,ma} = \sum_{s}^{S} \hat{w}_s \hat{\theta}_{j,s}$, where $\hat{w}_s$ is the weight for the $s$th model and $\hat{\theta}_{j,s}$ is the estimate of the $j$th parameter in the $s$th model. $\hat{\theta}_{j,s}$ is zero if the $j$th parameter is not included in the $s$th model. Then, the variance of the $j$th averaged parameter can be calculated as $\mathrm{var}\left(\hat{\theta}_{j,ma}\right) = \left(\sum_{s}^{S} \hat{w}_s \sqrt{\mathrm{var}\left(\hat{\theta}_{j,s}\right) + \left(\hat{\theta}_{j,s} - \hat{\theta}_{j,ma}\right)^2}\right)^2$, where $\mathrm{var}\left(\hat{\theta}_{j,s}\right)$ is the variance of the $j$th parameter in the $s$th model and is defined as zero if the $j$th parameter is not included in the $s$th model[27]. The composite likelihood is then maximized over the weights for the five models, where the initial weights for the optimization are based on a modified Akaike information criterion (AIC) for composite likelihood[28], for the five models (see Supplementary Note).

Finally, the averaged model will be selected as the final model if either of the following criteria is met: (1) the composite likelihood of the averaged model is higher than that of the $s_{1,2,C}$ model; or (2) for either of the two traits, the estimate of the trait-specific heritability from the $s_{1,2,C}$ model is <0.05 and the estimate of the corresponding trait-specific mixing proportion is not significant. Otherwise, the $s_{1,2,C}$ model is selected as the final model.

**Genetic correlation calculation**

Based on our reciprocal joint model, genetic correlation ($r_g$) between the two phenotypes is calculated as

$$r_g = \frac{\pi_1 \delta_{21} \sigma_1^2 + \pi_2 \delta_{12} \sigma_2^2 + \pi_C \left[\delta_{21} \sigma_{C1}^2 + \delta_{12} \sigma_{C2}^2 + (1 + \delta_{12}\delta_{21})\rho_{C1,C2}\right]}{\sqrt{\left(\pi_1 \sigma_1^2 + \pi_2 \delta_{12}^2 \sigma_2^2 + \pi_C \left[\sigma_{C1}^2 + \delta_{12}^2 \sigma_{C2}^2\right]\right) \cdot \left(\pi_1 \delta_{21}^2 \sigma_1^2 + \pi_2 \sigma_2^2 + \pi_C \left[\delta_{21}^2 \sigma_{C1}^2 + \sigma_{C2}^2\right]\right)}}$$

The derivation of $r_g$ and the corresponding variance calculation can be found in Supplementary Note.

**Simulation data**

In summary, the simulation data were generated using imputed genotypes of 50,000 individuals randomly extracted from UK Biobank white British subjects. We only selected common SNPs (i.e., MAF ≥ 0.05) that are available in the HapMap 3 reference panel, resulting in a total of about 0.68 million SNPs. The majority of the simulations were based on 100% overlapping samples with a sample size of 50,000. Moreover, we tested our model using 0 and 50% overlapping samples with a sample size of 50,000. We also tested our model using 100% overlapping samples with a sample size of 20,000. The phenotypes were simulated using GCTA software[29] based on these genotype data and the association tests were performed using PLINK software[30]. Detailed simulation scenarios (represented by simID in the table) and corresponding parameter settings can be found in Supplementary Table 1.

Specifically, in each simulation scenario, we first defined values of $\pi_1, \pi_2, \pi_c, \sigma_1^2, \sigma_2^2, \sigma_{C1}^2, \sigma_{C2}^2, \rho_{C1,C2}, \delta_{12}$ and $\delta_{21}$ for the scenario. The number of causal SNPs in each component was determined according to the

assigned mixing proportions for the four SNP components. Then, the corresponding number of causal SNPs was randomly chosen from the set of available SNPs. For trait-specific SNPs, the direct effect sizes for causal SNPs were assigned based on $\gamma_{1i} \sim N(0, \sigma_1^2)$ and $\gamma_{2j} \sim N(0, \sigma_2^2)$ with $i \in G_1$ and $j \in G_2$. For pleiotropic SNPs, the direct effect sizes for causal SNPs were assigned based on $\begin{pmatrix} \gamma_{C1l} \\ \gamma_{C2l} \end{pmatrix} \sim N\left[\begin{pmatrix} 0 \\ 0 \end{pmatrix}, \begin{pmatrix} \sigma_{C1}^2 & \rho_{C1,C2} \\ \rho_{C1,C2} & \sigma_{C2}^2 \end{pmatrix}\right]$, with $l \in G_C$. Next, we converted the direct effect sizes to the component-dependent joint effect sizes using $\boldsymbol{\beta}_k^{(h)} = [\boldsymbol{I} - \boldsymbol{\Delta}]^{-1} \boldsymbol{\Gamma}_k^{(h)}$ to take into account reciprocal causation. In this way, we could obtain the joint effect sizes of causal SNPs for each phenotype. Then we simulated the two phenotypes ($Y_1$ and $Y_2$) with these joint effect sizes using GCTA software and performed the association test using PLINK to obtain the summary-level results. Finally, we repeated the simulations 100 times for each scenario.

For the full-model scenario ($S_{1,2,C}$), we generated data under three causation scenarios: "bi-directional causation" when there are causal effects in both directions, "uni-directional causation" when there is a causal effect in only one direction, and "null causation" when there is no causal effect in either direction. We considered low and high polygenicity scenarios. For low polygenicity scenarios (LoS1–LoS12), we performed two sets of simulation studies: (I) independent pleiotropy ($\pi_1 = \pi_2 = \pi_c = 1 \times 10^{-4}$ and $\rho_{C1,C2} = 0.0$); and (II) correlated pleiotropy ($\pi_1 = \pi_2 = \pi_c = 1 \times 10^{-4}$ and $\rho_{C1,C2} = 0.1$) (Fig. 1b). For high polygenicity scenarios, we set $\pi_1 = \pi_2 = \pi_c = 1 \times 10^{-3}$ and performed six simulation studies under correlated pleiotropy considering bi-directional, uni-directional and null causations (HiS1–HiS6). We also additionally simulated a high polygenicity scenario in which the two phenotypes had different mixing proportions and genetic variances (HiS7). For simulation studies with different sample sizes, we set $\pi_1 = \pi_2 = \pi_c = 1 \times 10^{-3}$ and decreased the sample sizes to 20,000 under correlated pleiotropy scenarios (SS1–SS3).

For simulations involving binary phenotypes (HiS8), we assumed $Y_1$ to be the continuous phenotype and $Y_2$ to be the binary phenotype. For $Y_1$ we used the above method (with 50,000 individuals) to obtain GWAS summary data. For $Y_2$, we generated the joint effect sizes on the liability scale, then took a total of 300,000 individuals to simulate a case-control study with a disease prevalence of 0.05 and case:control ratio of 1:2, using GCTA. Thus, we generated 15,000 cases from the 300,000 individuals and generated 30,000 controls from the remaining individuals. Consequently, the total sample size for the binary phenotype is 45,000. We then ran logistic regression analysis using PLINK to obtain the GWAS summary statistics for $Y_2$. Since the parameters in our model are on the liability scale, we converted the summary-level estimates of odds ratios to the liability scale effect size through approximation[31,32] (see Supplementary Note).

For sub-model scenarios, we generated three sets of simulated data: no $Y_1$-specific component ($S_{2,C}$), no pleiotropy component ($S_{1,2}$), and pleiotropy component only ($S_C$). For each sub-model, we further simulated data for bi-directional, uni-directional and null causations.

Moreover, we designed simulations using unbalanced effects for genetic components, which means that the genetic effects for each component are different. Under these scenarios, we first simulated data with sample overlapping changing from 0% to 100% (Unbalance 1–3). Then, we simulated sub-model scenarios with unbalanced settings (Unbalance $S_{1,2,C}$, $S_{2,C}$, and $S_C$).

Representative bivariate scatterplots for these simulations can be found in Fig. 1 and Supplementary Fig. 1.

**Existing MR methods and LD score regression**

We also ran several IV-based MR methods on the simulated data: Egger regression, Weighted Median, Weighted Mode, Inverse-Variance Weighted (IVW) (from the TwoSampleMR package[3]), MRMix[12], and MR-PRESSO[9]. We tested three SNP selection methods: (a) use valid IVs,

i.e. all the exposure-specific true causal SNPs assigned in the simulation as IVs; (b) use significant exposure-associated SNPs ($p$-value $< 5 \times 10^{-8}$) but exclude potential outcome-associated SNPs by setting $p$-value $> 5 \times 10^{-5}$ in outcome GWAS; (c) use significant exposure-associated SNPs regardless of their association with outcome. For SNPs selected in (b) and (c), we further performed clumping ($r^2 < 0.01$) to obtain the independent IVs for MR analysis. Additionally, we also compared with CAUSE[13], a recent method using genome-wide summary statistics. The $p$-value thresholds for CAUSE and MRMix were set as $1 \times 10^{-3}$ and $5 \times 10^{-8}$, respectively. To calculate the genetic correlation using LD Score regression[1], we followed the online tutorial and used the default settings.

### Real data processing

We followed steps similar to those implemented by Zhang et al.[26] to preprocess the public GWAS summary data. Briefly, SNPs were excluded if MAF was <5%; or if the imputation INFO score was low (INFO < 0.9); or if the available sample size was less than 0.67 of the 90th percentile of the available sample sizes for all SNPs; or if it was located within the major histocompatibility complex (MHC) region; or if the absolute value of standardized effect size is >0.1. The summary statistics of the remaining SNPs were merged with 1000 Genomes Project reference SNPs to obtain their corresponding LD scores.

### Data availability

The genotype data for simulation were applied from UKB biobank (www.ukbiobank.ac.uk). LD infomation data were from GENESIS (https://github.com/yandorazhang/GENESIS) with permission. Public GWAS summary data were from the corresponding website: MAGIC consortium (ftp://ftp.sanger.ac.uk/pub/magic); CARDIoGRAMplusC4D (http://www.cardiogramplusc4d.org/data-downloads); Global Lipids Genetics Consortium (http://csg.sph.umich.edu/willer/public/lipids2013/); DIAGRAM consortium (https://diagram-consortium.org/downloads.html); EGG Consortium (http://egg-consortium.org/); GIANT consortium (http://portals.broadinstitute.org/collaboration/giant/index.php/Main_Page); GWAS catalog (https://www.ebi.ac.uk/gwas/downloads/summary-statistics).

### Code availability

The MRCI program code can be found at https://github.com/zpliu/MRCI. Detailed results of all real data analyses can be visualized through the website: https://triangularcell.shinyapps.io/MRCI_Estimate_for_CommonDiseases.

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

## Acknowledgements

We thank the consortia for sharing the GWAS summary statistics. We thank Dr. Robert Porsch for critical advice on the model. The simulation studies were conducted using genotype data from the UK Biobank Resource accessed under Application Number 28732. The computations were performed using research computing facilities offered by Information Technology Services, The University of Hong Kong, and the High-Performance Computing Facility of Bioinformatics Core, Centre for PanorOmic Sciences (CPOS), LKS Faculty of Medicine, The University of Hong Kong. This work was supported by Hong Kong Research Grants Council Collaborative Research Grant C7044-19G, Hong Kong Innovation and Technology Bureau funding for the State Key Laboratory of Brain and Cognitive Sciences, and the National Natural Science Foundation of China (32170637).

## Author contributions

P.C.S., Y.D.Z., Z.L., and Y.Q. conceived and designed the model and contributed to the interpretation of the results. Z.L. and Y.Q. designed the algorithm, implemented the software, and conducted analysis of simulation and real data. T.S.H.M. prepared the genotype data for simulation. T.W. contributed to statistical derivation. Y.D.Z. and M.L. prepared LD data and developed the software. L.B. and J.D.T. contributed to the interpretation of real data analysis. P.C.S., Z.L., and Y.Q. contributed to writing the manuscript. T.W., J.D.T., L.B., M.L., and Y.D.Z. contributed to the critical revision of the manuscript.

## Competing interests

The authors declare no competing interests.
