## [Transparent Peer Review File · Nature Communications]

Reciprocal causation mixture model for robust mendelian randomization analysis using genome-scale summary dataWe really appreciate that the reviewers offered useful comments to improve our manuscript. Here, we have prepared a point-by-point response in blue.

REVIEWER COMMENTS

Reviewer #1 (Remarks to the Author: Overall significance):

Liu et al propose a normal mixture model for the genetic effects on two traits and include bidirectional causal effects that can be estimated under Mendelian randomisation assumptions allowing for both correlated and uncorrelated pleiotropy. The method is shown to work well under simulations and improve upon standard and related methods including CAUSE and MR MIX. Although the authors are not the first to propose this type of model, there are some very nice innovations in this work, including the composite likelihood with sandwich variance to allow for LD, and the model averaging to allow for missing components.

Thanks for the summary.

A very similar method has been on medRxiv for a couple of years now, the most recent version here <https://www.medrxiv.org/content/10.1101/2020.01.27.20018929v2.full>. I would not demand discussion of a work that has yet to complete peer review, but the authors should be honest if they have drawn upon this related work at all.

Thanks for this information. First, we finished the major part of our model independently before this paper was published on medRxiv. Indeed, we presented our model as a poster at the 30th International Conference on Genome Informatics in Sydney on December 9-11, 2019, but the paper in medRxiv was not published until Jan 27, 2020. We did subsequently notice the paper, but did not discuss it in our submitted manuscript mainly because when we first submitted our manuscript (Jul 19, 2021), this work had not yet completed peer review.

Reviewer #1 (Remarks to the Author: Impact):

Altogether this looks like a substantive and useful contribution.

Reviewer #1 (Remarks to the Author: Strength of the claims):

1. The authors performed a lot of simulations and I am not one to quibble over details. However there are a few overarching issues.

a. Only the one-sample design was simulated. This creates weak instrument bias towards the observational association, which was observed in the simulations, but in the two-sample design the bias is towards the null. Given the popularity of the two-sample design, there should be some simulations comparing MR CI to standard methods in this case.

Thanks for the suggestions. In the revised manuscript, we added simulation for a two-sample design. In these two-sample simulations, we used the genotypes of 50K non-overlapping individuals (from UK Biobank) to simulate each phenotype (Supplementary Table S1). The causal effects were set as $\delta_{12} = 0.1$ and $\delta_{21} = 0.0$. We compared the estimates from our model with those from several existing MR methods. The results showed that the estimates of causal effects in both directions were still nearly unbiased. Additionally, the power to detect the effect in the non-null causal direction (δ_{12}) was adequate and the type I error rate for the null causal direction (δ_{21}) was well-controlled

(Supplementary Figure S3a-b, as shown under comment No.1b.).

b. Furthermore, since the model allows partial sample overlap, there should be some simulations showing that the method works correctly in this situation.

Thanks for the suggestion. In the revised manuscript, we added simulation for partial sample overlap. In these partial-overlap simulations, we used 50K individuals for simulation of the sample for each phenotype, and 50% of the individuals (i.e. 25K individuals) were overlapping between the two samples (Table below). The causal effects were set as $\delta_{12} = 0.1$ and $\delta_{21} = 0.0$. The results showed that estimates of both causal effects were still unbiased. Additionally, the power to detect the effect in the non-null causal direction (δ_{12}) was also adequate and the type I error rate for the null causal direction (δ_{21}) was well-controlled (Supplementary Figure S3a-b, as shown below). Our method includes a sample overlap parameter ρ_0 , and we observed that estimates of ρ_0 increased as the sample overlap increased in the simulations (Supplementary Figure S3c). This supports our method's ability to handle sample overlap in a reasonable manner. This might be an important advantage as more biobank-scale data (in addition to the UK Biobank) incorporating multiple phenotypes become available in the future.

Table. Simulation settings for different sample overlaps.

simID	π_1	π_2	π_c	h_1^2	h_2^2	h_{c1}^2	h_{c2}^2	$\rho_{c1,c2}$	δ_{12}	δ_{21}	sample size (Y_1/Y_2)	sample overlapping
Unbalance1	5e-4	2e-3	5e-3	0.2	0.3	0.1	0.2	0.1	0.1	0.0	50K / 50K	0%
Unbalance2	5e-4	2e-3	5e-3	0.2	0.3	0.1	0.2	0.1	0.1	0.0	50K / 50K	50%
Unbalance3	5e-4	2e-3	5e-3	0.2	0.3	0.1	0.2	0.1	0.1	0.0	50K / 50K	100%

Supplementary Figure S3. Estimation comparison using different methods under various sample overlap conditions. Data were from simulations with 0%, 50% and 100% sample overlap, respectively. The causal effects were set as: $\delta_{12} = 0.1$ and $\delta_{21} = 0.0$. **a**, shows causal estimates from MRCI and selected standard MR methods. MRCI shows nearly unbiased estimates regardless of sample overlap. **b**, type I error rate (δ_{21}) and power (δ_{12}) of detecting causal effects by MRCI and selected existing MR methods. MRCI shows adequate power and correct type I error rate under different sample overlap scenarios. **c**, estimates of nuisance parameter ρ_0 for different degrees of sample overlap, showing that the estimates of ρ_0 increase as the degree of sample overlap

increases. In the simulation, the mixing proportions of π_1 , π_2 and π_C were set as 5×10^{-4} , 2×10^{-3} and 5×10^{-3} respectively; the heritabilities of h_1^2 , h_2^2 , h_{C1}^2 , and h_{C2}^2 were set as 0.2, 0.3, 0.1 and 0.2, respectively; $\rho_{h_{C1}^2, h_{C2}^2}$ was set as 0.1.

c. Similarly, since the model allows for population stratification, there should also be some corresponding simulations.

Population stratification can certainly inflate GWAS summaries. However, most reported GWAS will have performed appropriate correction, such as principal components and genomic control, to minimize population stratification effects on association test statistics. Thus, population stratification should not be a critical issue for published GWAS summary data. This was why we did not perform simulations with uncorrected population stratification. Nevertheless, the parameters for the population stratification are considered as nuisance parameters and are used as an extra safeguard.

d. The authors observed increased bias when IVs are selected according to sample associations with exposure. This looks like a winner's curse effect, which is a different problem to pleiotropy and weak instrument bias. Since this is often done in practice, the authors are welcome to demonstrate the effect, but it would also be helpful to see what happens when IVs are selected from an independent sample, as done eg in Zhao Q et al Int J Epidemiol 2019. Although the present results using "true" IVs get at this somewhat, it would still help if the authors clarified the presence of winner's curse effect.

Thanks for this suggestion. First, our model does not have to select instruments for estimation as it does not use a classic MR framework. Instead, we consider all available SNPs of the two GWAS and using a mixture model. Second, according to Zhao Q *et al*, the presence of the 'winner's curse' generally biases the causal effect estimate towards zero. However, we observed an upward bias in the estimates for MR methods such as IVW in our simulation (Figure 2a). Moreover, when we removed pleiotropy effects in simulations, the estimates from these classical MR methods became nearly unbiased and the type I error rate was well-controlled (Figure below). Thus, the bias is likely to be caused by pleiotropic SNPs rather than the winner's curse effect. In practice, for many complex traits, it might be difficult to find a third GWAS, which should have adequate power for accurate estimation of SNP effects, to obtain instruments.

Figure. Estimates from models with no-pleiotropy and correlated-pleiotropy. When pleiotropy effects were absent, all methods showed unbiased estimates and correct type I error rates. Thus, the overestimates under correlated pleiotropy scenarios for some MR methods are unlikely to be caused by the winner’s curse effect. In the simulation, $\delta_{12} = \delta_{21} = 0.0$, $\pi_1 = \pi_2 = 0.001$ and $h_1^2 = h_2^2 = 0.3$. For the no-pleiotropy simulations, $\pi_C = h_{C1}^2 = h_{C2}^2 = \rho_{h_{C1}^2, h_{C2}^2} = 0.0$. For the correlated-pleiotropy simulations, $\pi_C = 0.001$ and $h_{C1}^2 = h_{C2}^2 = \rho_{h_{C1}^2, h_{C2}^2} = 0.1$.

e. In the time-honoured way, the authors’ method performed better than its competitors and recovered the desired statistical properties. When would it do less well? What assumptions does it require, that other methods may not? In particular, why does MRCI do better than CAUSE and MRMIX, when the models are so similar? What is creating the improvement? Why does CAUSE have increased type-1 error? Insights are welcome.

Thanks for these comments. First, we have to admit that our model is not perfect in every way. We have discussed limitations for our method in the manuscript (as follows):

“Our assumption of bivariate normality of standardized marginal effects in the four SNP components may not hold in practice, and serious violation of this assumption may affect the performance of the method. Further simulations to evaluate the sensitivity of the method to violations of this assumption are desirable. A second limitation is that we used composite likelihood for model fitting

because marginal SNP effects are correlated. This necessitated the use of robust sandwich standard errors, which may lead to conservative, less powerful tests. Finally, in the scenario where pleiotropic effects are very strong and nearly no causal SNPs contribute specifically to the exposure, MRCI will typically produce very large standard errors for the causal path estimate from exposure to outcome, so that there is little power to detect such a causal effect even when it is present. However, this situation of having almost no SNPs that can be used as valid IVs could be a limitation for the MR approach in general, rather than for MRCI specifically.”

Second, in terms of model design, CAUSE and MRMix are similar to our full-model in some ways, e.g., allowing correlated pleiotropy in the model. In terms of estimation performance, CAUSE and MRMix are also comparable to MRCI using our full-model simulation. However, when it comes to sub-model scenarios, they typically produced biased estimates of causal effects and inflated type I errors. Thus, these methods are vulnerable when a particular component in the model is missing. Additionally, there are also differences in model implementation: a major difference is that MRCI included both SNP effects and LD information during estimation (please see Methods for detail) while CAUSE transformed the SNP effects using LD, and MRMix used independent SNPs.

2. Page 6, “estimation assuming the full model often did not correctly infer the absent component”. I wondered whether this might be as result of the numerical maximisation when the mixing proportions must be within [0,1]. Optimisation can work better when such variables are parameterised on the logit scale and I wondered whether this was done. Similar comments apply to the variances.

Thanks for your suggestion. In our current algorithm, we did not perform logit transformation for these parameters. Indeed, numerical maximisation within a restricted range could be an issue for an Expectation–Maximization algorithm. We think this kind of restriction is necessary for both theoretical and practical purpose. From our simulations optimization is sometimes problematic, and we may consider this suggestion in the future.

3. In the data examples, some of the inferred effects are implausible, such as T2D on birth weight. Also CAD on BMI and T2D on BMI are doubtful: although indeed these diseases can lead to weight loss, what the data show is that genetic predisposition to these diseases is negatively correlated with BMI in the general population (because the BMI effects are taken from population studies). And because most people in the BMI studies do not have CAD or T2D, we are probably not seeing causal effects of the diseases themselves. What aspects of this model are leading to implausible inferences?

Thanks for this comment. First, the model assumes that both phenotypes are polygenic, and defines SNP effects on binary phenotypes (such as T2D) to be on the liability scale. Thus, the causal estimates for binary traits are also on the liability scale. Second, in order to check the power of our method when a binary trait shows low prevalence in population, we simulated a binary trait with a prevalence of 1% and simulated the other continuous trait. The parameters in simulation were obtained from the T2D-BMI estimation results (as shown in the table below). Results from 100 simulations showed that MRCI could recover the causal effects using a prevalence of 1% (Figure below). Power for these estimates is 0.81 and 0.88 for δ_{12} and δ_{21} , respectively. These results support the ability of the model to detect a causal effect of a disease, even when the disease is present in only a minority of the GWAS on the other phenotype.

Table. Simulation settings to recover BMI-T2D results. In simulation, Y_1 is a continuous trait and Y_2

is a binary trait. The genotype data were from individuals in UK Biobank. The prevalence and case:control ratio for Y_2 were set as 1% and 1:9, respectively. h^2 values for Y_2 were defined on a liability scale. Sample sizes and case:control ratio were determined based on the reported information from the corresponding publications of BMI (Yengo L, *et al. Hum Mol Genet.* 2018) and T2D (Mahajan A, *et al. Nat Genet.* 2018) GWAS.

π_1	π_2	π_c	h_1^2	h_2^2	h_{c1}^2	h_{c2}^2	$\rho_{c1,c2}$	δ_{12}	δ_{21}	Sample size (Y_1 / Y_2)
6.18e-3	4.92e-4	1.74e-4	0.17	0.09	0.027	0.018	0.001	-0.17	0.7	300K / 300K

Figure. Estimates from 100 BMI-T2D simulations. True values for δ_{12} and δ_{21} are shown in dashed lines.

4. As seen in figure 1, the model is perfectly symmetric. This suggests that there are (at least) two solutions that fit any data set equally well. How do we deal with this?

Thanks for this comment. We apologize that we did not make this point clear. Our model is perfectly symmetric in the sense that which of the two phenotypes (A or B) is assigned as Y_1 is entirely arbitrary. However, once the phenotypes have been assigned, the genetic parameters of the two phenotypes and causal effects in the two opposite directions would be interpreted accordingly. Whether A or B is assigned as Y_1 should produce the same fitted model, when the parameters are interpreted correctly according to the assignment.

To further illustrate this point, we simulated two phenotypes A and B with the same genetic component parameters ($\pi_{A-specific} = \pi_{B-specific} = \pi_{pleiotropy} = 1 \times 10^{-4}$; $h_{A-specific}^2 = h_{B-specific}^2 = 0.3$; $h_{pleiotropy-A}^2 = h_{pleiotropy-B}^2 = 0.1$; $\rho_{h_{pleiotropy-A}^2, h_{pleiotropy-B}^2} = 0.1$), but with two different phenotypic causal effects in the two directions ($\delta_{B \rightarrow A} = 0.1$ and $\delta_{A \rightarrow B} = 0.05$). We ran two separate analyses on each of such simulated data sets: for the first analysis, we set A as Y_1 and B as Y_2 ; for the second one, we set A as Y_2 and B as Y_1 . The results showed that our method correctly estimated the causal effects in both direction regardless of the assignment of Y_1 and Y_2 (see Table below).

Table. Estimates after swapping input phenotypes for Y_1 and Y_2 . Results were summarized from 100 simulations.

Assignment	Causal direction	True value	Mean estimate	SD
$Y_1 = A; Y_2 = B$	$A \rightarrow B$	0.05	0.053	0.016
	$B \rightarrow A$	0.1	0.098	0.019
$Y_1 = B; Y_2 = A$	$A \rightarrow B$	0.05	0.051	0.019
	$B \rightarrow A$	0.1	0.096	0.021

5. The sandwich variance is a very nice idea, but it does require a large sample. Is it possible to give some guidance on the minimal sample size where this approach is reasonable?

Thanks for your comment. Sandwich variance is a common idea for parameter variance estimation when using composite likelihood (Varin C. *AStA Advances in Statistical Analysis*. 2008). As the estimation accuracy also depends on the particular genetic architecture of the phenotype, it is difficult to give a precise guidance on the minimal sample size. In simulations we tried a relatively small sample size of 20K for both phenotypes, and our method still performed reasonably well (Figure 4C).

6. Page 13, unclear how the case/control data were simulated. “45,000 individuals to simulate a case-control study with disease prevalence 0.05 and case:control ratio 1:2”, does this mean there were $0.05 \times 45k = 2,250$ cases and twice as many (4500) controls? Or does it mean 15,000 cases and 30,000 controls, in which case a much larger number of individuals must be simulated to obtain that many cases.

Thanks for pointing out this issue. Sorry for the confusion here. We used a total of 300,000 individuals to simulate a case-control study with a disease prevalence of 0.05 and case:control ratio 1:2, using GCTA. Thus, we generated 15,000 cases from the 300,000 individuals and generated 30,000 controls from the remaining individuals. So the actual sample size used for this case-control study is 45,000 individuals. We have clarified this in the revised manuscript.

7. Please check referencing, eg ref 26 is missing authors and publisher, ref 28 volume and page numbers.

Thanks for pointing out this issue. Ref 26 is text book, so the citation format is different. For ref 28, we have revised the citation in the revised manuscript.

8. Supp text, top page 5, should BIC read AIC?

Thanks for pointing out this typo. It should read AIC, and we have revised this in the revised manuscript.

9. Could the authors please make their software available?

Actually, we have already provided the github link in the submitted manuscript under ‘Code availability’ section. According to the Journal’s guideline, we will keep the link under this section in

the revised manuscript. And for your convenience, the MRCI method can be found at <https://github.com/zpliu/MRCI>

REVIEWER COMMENTS

Reviewer #2 (Remarks to the Author: Overall significance):

The proposed approach MRCI mainly focused on handling potentially correlated horizontal pleiotropy which is an essential but challenging task in Mendelian randomization. It is an interesting and solid work in general. The authors provided a methodologically robust framework and demonstrated its utility by simulations assuming strongly correlated pleiotropy effects. However, as the authors pointed out, the hypothesis of a strongly correlated pleiotropy effect needs to be justified in the further with evidence from empirical datasets. The feature of the "two-way test in one run" is interesting but a bit trivial since it won't take much time to run regular MR on the other way when the GWAS datasets have been harmonized.

Thanks for the summary. We agree that the hypothesis of strong correlated pleiotropy remains to be empirically tested, even though moderate to high genetic correlations have been found between many phenotypes (Bulik-Sullivan B, *et al. Nat. Genet.*, 2015; Verbanck M, *et al. Nat. Genet.* 2018; Anttila V, *et al. Science.* 2018). However, if pleiotropy reflects common biological mechanisms underlying multiple phenotypes, then it would be surprising for pleiotropic effects to be uncorrelated. We, as well as other experts in the field of MR studies, are well aware that the correlated pleiotropy is a potential source of false causal inference (O'Connor L. J. & Price A. L. *Nat. Genet.* 2018; Jean Morrison, *et al. Nat. Genet.* 2020). We also agree with the reviewer that running regular MR analysis in two directions is straightforward as long as we are able to find valid instruments. However, in practice, it is not always easy to identify valid instruments for both directions. Thus, being able to do a 'two-way test in one run' without having to select valid instruments may still be an attractive feature, though not a very important one.

Reviewer #2 (Remarks to the Author: Impact):

Included above.

Reviewer #2 (Remarks to the Author: Strength of the claims):

Non-technical comments:

1. It is hard to write comments without line numbers and page numbers.

Line numbers and page numbers have been added in the revised version.

2. Please include figures and figure legends in the main text during peer-review. Otherwise, the reviewers need to jump back and force three different places which is annoying.

These have been added into the main text in the revised version.

3. The authors may also notice the MR approach "MRCIP" published very recently. Both "MRCI" and "MRCIP" target on correlated pleiotropy effect. What a coincidence. I suggest the authors find another name since people may be confused about them (e.g., consider MRCIP as an extension of MRCI).

Thanks for this information. This MRCIP stands for "MR approach accounting for correlated and

idiosyncratic pleiotropy” and our MRCI stands for “Mixture model Reciprocal Causation Inference”. We were not aware of this MRCIP when we wrote our manuscript, thus, this one is really a coincidence to us. We think that “Mixture model Reciprocal Causation Inference” is the best summary for our method, so we would like to stick to “MRCI”.

Technical comments:

1. The second paragraph of the introduction needs to be reorganized. The summary of the MR history is incomplete.

Thanks for this comment. Please check the expanded ‘Introduction section’ in the revised manuscript.

The logic is not clear by saying “However, using multiple SNPs as IVs also increases the chance of horizontal pleiotropy for some SNPs” given the context that they just mentioned “weighted median” and “weighted mode”.

We really appreciate that the reviewer has pointed out this. Using multiple SNPs could increase the power of MR inference as long as the SNPs satisfy the three classical MR assumptions. However, in reality, it is not always an easy process to pick out a large number of valid SNPs from the others, as pleiotropy is likely to be widespread among complex traits (Verbanck M, *et al. Nat. Genet.* 2018). By including more SNPs, it might also increase the chance of including SNPs with pleiotropic effects. We have revised this as follows: “However, using multiple SNPs as IVs also increases the chance of horizontal pleiotropy for some SNPs, which would violate the assumptions for methods like inverse-variance weighting (IVW). Methods such as MR-Egger and MR-PRESSO address the issue of horizontal pleiotropy under the InSIDE (instrument strength independent of direct effect) assumption. Methods such as Weighted Median and Weighted Mode are robust to horizontal pleiotropy provided that there is an adequate proportion of valid IVs among the selected SNPs.”

The authors highlighted CAUSE as a recent advance after the sentence “... they still involve the selection of independent IVs, which may exclude the majority of SNPs”. However, CAUSE uses pruned/independent IVs.

We highly appreciate the reviewer for highlighting this point. CAUSE did not select independent IVs as classical MR methods. Instead, CAUSE makes use of whole-genome scale SNPs to calculate nuisance parameters. To fit a CAUSE model, however, an LD pruned set of variants are used to estimate posterior distributions (https://jean997.github.io/cause/ldl_cad.html). We have clarified this in the revised manuscript as follows: “One recent advance is to account for correlated pleiotropy by estimating the nuisance parameters of a mixture model from randomly selected genome-wide summary data. This method still requires a set of linkage disequilibrium (LD) pruned variants to calculate posterior distributions to fit the causation model.”

From my understanding, the point to develop MRCI is providing a robust solution when the proportion of IVs with pleiotropy effect is much higher than its setting in CAUSE. CAUSE already provided a way to address correlated pleiotropy. The authors need to highlight it in the background if they agree with this point.

Thanks for this comment. First, we agree that CAUSE already provides a way to address correlated

pleiotropy, which can be found in results under certain full-model simulations. Second, our method is more robust when one or two SNP components are missing in the model (i.e., sub-model scenario). Under such sub-model scenarios, CAUSE would show biased estimates and inflated type I error, probably due to violation of its model design.

2. The authors need to give a clear definition for what does “correlated pleiotropy” mean in the main text. It probably means correlated γ_{C1} and γ_{C2} in Figure 1. However, in the MR field, people are more familiar with the idea of the “violation of the InSIDE assumption”. I think the “correlated pleiotropy” and “violation of the InSIDE” are consistent when considering the G_c (Figure 1) as heritable Unobserved confounding factors. Readers will appreciate it if the authors could use established concepts.

Thanks for this suggestion. In our method, we explicitly modelled a component of pleiotropic SNPs with effect sizes represented by γ_{C1} and γ_{C2} . Thus, using “correlated pleiotropy” to represent the correlation between γ_{C1} and γ_{C2} may be more straightforward than using “violation of InSIDE assumption” to refer to heritable unobserved confounding factors.

3. In the first paragraph of “Estimation and hypothesis testing under the full model”, I don’t agree with the claim “However, IV-based MR methods generated biased estimates when GWAS significant SNPs for the exposure phenotype were used as IVs, especially under scenarios with correlated pleiotropy” based on the performance of “weighted mode”.

Thanks for pointing this out. We have revised the description to “many IV-based MR methods (except for Weighted Mode) generated biased estimates when GWAS significant SNPs for the exposure phenotype were used as IVs, especially under scenarios with correlated pleiotropy”. In fact, estimation power of Weighted Mode is lower than MRCI and other methods (Figure 2d).

4. Same paragraph as mentioned above, I think it is good to include CAUSE as a player in Figure 2. Although the authors made the comparison between CAUSE and MRCI in the following section.

Thanks for this suggestion. In Figure 2, in order to show that the selection of only significantly exposure-associated SNPs will bias the causal estimates when correlated pleiotropy effects appear, we only selected MR methods which require such a selection process. CAUSE does not select such SNPs, thus, we did not consider it appropriate to include CAUSE in this figure.

5. In the second paragraph of “Estimation and hypothesis testing under the full model”, please provide more details about “simulations with correlated pleiotropy” here.

We apologize that we did not make this point clear. Correlated pleiotropy means that the effects of pleiotropic SNPs on the two phenotypes are correlated. To reflect this correlation in simulation, we first defined a set of pleiotropic SNPs (the mixing proportion is represented as parameter π_c); then we defined the per-SNP genetic effects of these pleiotropic SNPs to have variances σ_{C1}^2 and σ_{C2}^2 , for phenotype Y_1 and Y_2 , respectively. The covariance between these pleiotropic effects was likewise defined as $\rho_{C1,C2}$. In this way, we defined correlated pleiotropy when $\rho_{C1,C2} \neq 0$.

6. Restrictions need to be mentioned in the first sentence of the second paragraph of “Estimation

and hypothesis testing under the full model". e.g., by assuming extensive correlated pleiotropy effect.

Thanks for this comment. The results in this paragraph were from both independent and correlated pleiotropy. Please check Supplementary Table S1 for details of simulation settings of various independent and correlated pleiotropy scenarios (simID LoS1-LoS12). Please also check Figure 2a-c, Supplementary Figure S2 and Supplementary Table S2-S7 for the results of the corresponding scenario.

7. Sections "Estimation and hypothesis testing under sub-models" and "Comparison with CAUSE and MRMix". Although I can certainly see the flexibility of the approach from a methodological perspective (which is great), it is very hard for me to picture a real scenario that trait-specific SNPs are absent for one or both phenotypes. Discusses about the relationship between such extreme conditions and the observations of genetic correlations from empirical data is needed.

Thanks for this comment. From a methodological perspective, we agree that MRCI provides flexibility to deal with more "extreme" sub-model conditions. From a practical perspective, these "extreme" conditions may not be uncommon. Taking psychiatric disorders as examples, one study across eight psychiatric disorders (Cross-Disorder Group of the Psychiatric Genomics Consortium. *Cell*. 2019) has reported that nearly 75% of the genome-wide significant SNPs were pleiotropic (i.e., associated with more than one disorder). Indeed, it could be unrealistic for a trait-specific component to be completely absent. However, as found in psychiatric disorders, the trait-specific effects may be much smaller relative to the substantial pleiotropy effects. We admit that the power to detect a causal effect in our model is limited when pleiotropy effects dominate over trait-specific effects. We consider this to be preferable to greater power at the expense of false positive causal inference. If GWAS data are available for endophenotypes of psychiatric disorders, it would be interesting to investigate the causal relationship using endophenotypes.

8. The observation of a reciprocal causal relationship between BMI and T2D is interesting. I am trying to get my head around these biologically reciprocally correlated phenotypes. I am wondering whether using a GWAS of T2D_adj_BMI and a GWAS of BMI would give us a more intuitive picture of the causality from T2D to BMI.

We highly appreciate the reviewer for highlighting this very important point. First, the model assumes that both phenotypes are polygenic, and defines SNP effects on binary phenotypes (such as T2D) to be on the liability scale. Thus, the causal estimates for binary traits are also on the liability scale. Second, in order to check the power of our method when a binary trait shows low prevalence in a population, we simulated a binary trait with a prevalence of 1% and simulated the other continuous trait. The parameters in this simulation were obtained from the T2D-BMI estimation results (as shown in the table below). Results from 100 simulations showed that MRCI could recover the causal effects using a prevalence of 1% (Figure below). Power for these estimates is 0.81 and 0.88 for δ_{12} and δ_{21} , respectively. These results support the ability of the model to detect a causal effect of a disease, even when the disease is present in only a minority of the GWAS on the other phenotype. Additionally, as the reviewer has suggested, we also used a GWAS of T2D_adj_BMI (Mahajan A, et al. *Nat Genet*. 2018) and a GWAS of BMI (Yengo L, et al. *Hum Mol Genet*. 2018) to run the analysis. The reciprocal causations are still significant: the causal estimate from T2D to BMI is -0.13 (pval=5.83e-14) and the reverse causal estimate is 0.28 (pval=9.16e-21).

Table. Simulation settings to recover BMI-T2D results. In simulation, Y_1 is a continuous trait and Y_2

is a binary trait. The genotype data were from individuals in UK Biobank. The prevalence and case:control ratio for Y_2 were set as 1% and 1:9, respectively. h^2 values for Y_2 were defined on a liability scale. Sample sizes and case:control ratio was determined based on the reported information from the corresponding publications of BMI (Yengo L, *et al. Hum Mol Genet.* 2018) and T2D (Mahajan A, *et al. Nat Genet.* 2018) GWAS.

π_1	π_2	π_c	h_1^2	h_2^2	h_{c1}^2	h_{c2}^2	$\rho_{c1,c2}$	δ_{12}	δ_{21}	Sample size (Y_1 / Y_2)
6.18e-3	4.92e-4	1.74e-4	0.17	0.09	0.027	0.018	0.001	-0.17	0.7	300K / 300K

Figure. Estimates from 100 BMI-T2D simulations. True values for δ_{12} and δ_{21} are shown as dashed lines.

9. Another two similar approaches have been published online recently (listed below). Comments in the Discussion will help readers to find their best fit.

Xu, Siqi, Wing Kam Fung, and Zhonghua Liu. “MRCIP: a robust Mendelian randomization method accounting for correlated and idiosyncratic pleiotropy.” *Briefings in Bioinformatics* (2021).

Cheng, Qing, Tingting Qiu, Xiaoran Chai, Baoluo Sun, Yingcun Xia, Xingjie Shi, and Jin Liu. “MR-Corr2: a two-sample Mendelian randomization method that accounts for correlated horizontal pleiotropy using correlated instrumental variants.” *Bioinformatics* (2021).

Thanks for the information. MRCIP is a very recent method to deal with correlated pleiotropy. Although MRCIP has relaxed the classical MR assumptions (MRCIP does not require the InSIDE assumption or the existence of valid instrumental variables), it still only considers independent SNPs in the model. MR-Corr2 uses an orthogonal projection strategy that first decomposes the direct effects into two parts, linear and orthogonal, then reparametrizes the relationship between SNP effects on exposure and outcome. MR-Corr2 partitions the whole genome into LD blocks to deal with LD among genetic variants. Additionally, both methods were developed for two-sample design, which may suffer from the sample overlapping issue.

Reviewer #2 (Remarks to the Author: Reproducibility):

Included above.

Reviewer #3 (Remarks to the Author: Overall significance):

This paper presents a novel MR estimation method that claims to make use of genome-wide associations and SNPs in LD to estimate bi-directional causal effects. This method is potentially interesting and its links to existing methods discussed appropriately. However, not enough description of the simulations or application are given to fully assess the method.

Thanks for the summary. We have revised the description of simulation data and application in the Method section. Please check these sections of the revised manuscript.

Reviewer #3 (Remarks to the Author: Impact):

This paper provides a novel method that could influence and improve the methods available for pleiotropy robust MR analysis however the limitations of the method, scenarios in which it would be most relevant and implementation of the method need to be more fully explored for its benefit to the field to be fully achieved.

Thanks for these comments. We have included discussion on limitations of the method. Please check the revised manuscript.

Reviewer #3 (Remarks to the Author: Strength of the claims):

1. This paper claims to use genome-wide SNPs in the estimation however in the simulation SNPs that have no association with either phenotype appear to be excluded from the analysis. Is this also the case in the applied analysis? In this case what rule is used to determine whether or not SNPs should be included in the applied analysis?

We thank the reviewer for the comments. Actually, our method does not exclude 'SNPs that have no association with either phenotype' from dataset, neither in simulation nor in applied analysis. Instead a specific subsets of SNPs (one of the four SNP components in the mixture model) was introduced to account for such SNPs. We include the following description of how the real data were pre-processed:

"We followed steps similar to those implemented by Zhang *et al* (*Nature Genetics*. 2018) to preprocess the public GWAS summary data. Briefly, SNPs were excluded if MAF was less than 5%; or if the imputation INFO score was low ($INFO < 0.9$); or if the available sample size was less than 0.67 of the 90th percentile of the available sample sizes for all SNPs; or if it was located within the major histocompatibility complex (MHC) region; or if the absolute value of standardized effect size is greater than 0.1. The summary statistics of remaining SNPs were merged with 1000 Genomes Project reference SNPs to obtain their corresponding LD scores. "

2. From the description of the distribution of the SNP effects, it appears all of the pleiotropic effects are assumed to be balanced. Given the potential issues that arise around unbalanced pleiotropy this is an important limitation of the method that needs to be highlighted clearly. This is particularly true around the discussion of correlated pleiotropy which is often considered to mean unbalanced directional pleiotropy in the same direction on each exposure. A discussion of this limitation and ideally a simulation illustrating the effect of unbalanced pleiotropy on the method would strengthen the manuscript.

We highly thank the reviewer for highlighting this point. We made some clarifications here. First, we have provided an unbalanced pleiotropy simulation in our previous manuscript (simID HiS7). This

was for a full-model scenario, which meant all four SNP components were present in the simulation. The genetic effects for pleiotropic SNPs are unbalanced, with the pleiotropic SNPs contributing to more variance of phenotype Y_2 than phenotype Y_1 . We also listed the detailed simulation settings (Table 1 below) and summary of the causal estimates using the full model (Table 2 below).

Table 1. Simulation settings for HiS7 unbalanced scenario.

simID	π_1	π_2	π_C	h_1^2	h_2^2	h_{C1}^2	h_{C2}^2	$\rho_{C1,C2}$	δ_{12}	δ_{21}	sample size (Y_1/Y_2)	sample overlapping
HiS7	5e-4	2e-3	5e-3	0.2	0.3	0.1	0.2	0.1	0.1	0.05	50K / 50K	100%

Table 2. Causal estimates for HiS7 unbalanced scenario.

simID	Parameter	True value	Mean estimate	Empirical SD	Mean SandwichSE	Mean χ^2 (SD)	Power
HiS7	δ_{12}	1.00E-01	1.09E-01	2.51E-02	2.56E-02	21.19 (13.2)	0.99
	δ_{21}	5.00E-02	5.03E-02	2.92E-02	2.38E-02	6.85 (6.84)	0.52

Additionally, we also simulated unbalanced pleiotropy simulation under sub-model scenarios (Table 3 below). As shown in the figure below, our final estimates under $S_{1,2,C}$, $S_{2,C}$ and S_C scenarios were still around the true values, and the type I error rate was well-controlled. We also compared estimation between our method and CAUSE and MRMix. Similar to observations in the balanced pleiotropy sub-model scenarios, when one or two components were missing ($S_{2,C}$ and S_C), CAUSE and MRMix generated biased estimates.

Table 3. Simulation settings for unbalanced genetic components.

simID	π_1	π_2	π_C	h_1^2	h_2^2	h_{C1}^2	h_{C2}^2	$\rho_{C1,C2}$	δ_{12}	δ_{21}	sample size (Y_1/Y_2)	sample overlapping
$S_{1,2,C}$	5e-4	2e-3	5e-3	0.2	0.3	0.1	0.2	0.1	0.1	0.0	50K / 50K	0%
$S_{2,C}$	0.0	2e-3	5e-3	0.0	0.3	0.1	0.2	0.1	0.1	0.0	50K / 50K	0%
S_C	0.0	0	5e-3	0.0	0.3	0.1	0.2	0.1	0.1	0.0	50K / 50K	0%

Finally, we also admit that under some extremely unbalanced pleiotropy scenarios, say one where pleiotropy effects are much stronger for one phenotype than the other, the performance of our model may be affected. However, since the performance also depends on other parameters in the model, we could not provide a particular threshold for the strength of such unbalanced pleiotropy effects.

3. Correlated pleiotropy is potentially a significant problem for MR analysis so it is really good that

the authors consider this in their simulations, however have they considered a higher level of correlated pleiotropy? How sensitive is the method to higher levels of correlated pleiotropy?

Thanks for this comment. A higher level of correlated pleiotropy is a limitation for our method; we have clarified this limitation in the Discussion section. In our simulation, the pleiotropy-only scenarios (S_C) show a relatively high correlation between the two pleiotropy effects (Supplementary Figure S1). Our estimation is still robust in controlling the type I error rate in this situation. But it might be difficult to define a particular sensitivity threshold for the level of correlated pleiotropy since other genetic components also affect the performance of our model. However, according to your comment No.2, we have performed 'unbalanced pleiotropy' simulations to test the performance under different pleiotropy scenarios.

4. Have you considered how sensitive the estimation method is to differences in instrument strength or the level of pleiotropy associated with the SNPs for each exposure? It would be beneficial to discuss these limitations and simulate the effect of (particularly) different levels of pleiotropy on the different phenotypes.

Thanks for this suggestion. We have performed simulation under 'unbalanced pleiotropy' scenarios, i.e., the effects of pleiotropy are at different levels, to test the performance. The results are shown under comment No.2.

5. In the section on sub-models and model averaging; can the authors expand on how using the submodels makes their estimation robust and what it is robust to?

We thank the reviewer for raising this point. Model averaging was introduced because we found that causal inference under the full model was prone to false positive causal inferences, when one of the four SNP components was absent from the true model. We interpreted the problem as arising from the difficulties in fitting a model with four components to a scenario that contains only three components, since the components were only distinguished from each other by their variances and covariances of effect sizes on the two phenotypes. We found that by fitting submodels and performing model averaging, we were able to control the rate of false positive causal inferences in these scenarios. In other words, the robustness of our method is to control the inflated type I error rate in sub-model scenarios.

6. In the introduction I was confused by the introduction of MR CAUSE as a model that uses LD pruned SNPs immediately after the discussion of the benefits of using non-independent SNPs since the LD pruning required by MR CAUSE restricts the analysis to independent SNPs.

We apologize for this confusion. CAUSE did not select independent IVs, as in classical MR methods. Instead, CAUSE makes use of whole-genome scale SNPs to calculate nuisance parameters. However, in the next step of fitting the causal model, an LD pruned set of variants is used to estimate posterior distributions (https://jean997.github.io/cause/ld_cad.html). Thus, CAUSE considers non-independent SNPs at the beginning of analysis.

Reviewer #3 (Remarks to the Author: Reproducibility):

Although the analysis appears to be appropriate (given the caveats of the limitations that haven't been explored discussed above), neither the simulations or the applied analysis are sufficiently

described for reproducibility or full assessment of the method.

1. I didn't understand how the data for the simulation had been generated. The paper suggests that it has been selected from UK Biobank, but in that case what is simulated? The description needs to be expanded on to fully understand how the simulations have been conducted.

We appreciate the reviewer for pointing this out. For UK Biobank data, we only used the raw genotype data, while the phenotypes were simulated based on these genotype data using the GCTA tool. We now elaborated the simulation part with more details. Please check the description for the simulation in the revised manuscript.

2. There is very limited description of the data used in the applied analysis. More than just a reference is needed given that this data is being used to illustrate a novel method.

Thanks for this comment. In this manuscript, we focused on introducing a robust method to infer causations using whole-genome scale summary data. In the applied analysis, we selected complex diseases and risk factors with adequate sample sizes for GWAS. We have summarized this GWAS information in Table 1, and offered our comments on the significant causal relations detected using this model. Among the pairs of risk factors and complex traits, some causal relations are widely accepted, e.g. LDL-CAD causation. However, for most of the phenotype pairs, their causal relations may still be debatable at present.

We really appreciate that the reviewer offered useful comments to improve our manuscript. Here, we have prepared a point-by-point response in blue.

REVIEWER COMMENTS

Reviewer #1 (Remarks to the Author: Overall significance):

The authors have responded to my comments. There are however a few issues outstanding. I use the numbers from my original review.

1c. This is a curious response. You say that you did not validate the model for population stratification, because in practice it would not be needed. Well in that case, why include these parameters at all?! They complicate the model fitting and increase standard errors. If you want to include these parameters, you should show that they do what they are supposed to do. Otherwise, leave them out.

Thanks for the comments. To further illustrate the function of the population stratification parameters, we designed three simulation scenarios (Figure 1a):

- (1) we only used Caucasian individuals and run the association test;
- (2) we allowed 1% and 5% of the total sample sizes to be non-Caucasian individuals for Y_1 and Y_2 , respectively and run association test without adjusting for the top principal components (PCs);
- (3) we used the same cohort settings as (2) but run the association test adjusting for the top10 PCs (adjPC).

We found that estimates of LDSC intercept and genomic control (λ_{GC}) were both inflated in the mixture simulation and proportional to the degree of stratification settings. The inflation could be controlled after adjusting for the top PCs (Figure 1b). Estimates of stratification factors in our model behaved similarly to λ_{GC} and could reflect the degree of population stratification (Figure 1b). The estimates of the causal effects did not show obvious bias in all simulated scenarios (Figure 1c), but the type I error rate was inflated when population stratification effects were not adjusted. The inflated type I error rate could be further decreased to an acceptable level after adjusting for the top PCs during association test (Figure 1d).

Figure 1. Simulations under stratified population. The ‘Caucasian’ simulations only included Caucasian individuals while the ‘mixture’ simulations included 1% and 5% non-Caucasian individuals for Y_1 and Y_2 , respectively. **a**, Representative scatterplots of the Caucasian and the mixture simulations. For the mixture simulations, we compared the summary statistics with or without adjusting for the top 10 principal components (PCs). **b**, estimates of LDSC intercept, genomic control (λ_{GC}) and estimates of stratification factors in our model were shown in the plots. The stratification factor parameters in our model reflected the increased level of population stratification and behaved similarly to the other two indices. **c**, Causal estimates of our method in stratified simulations were still near the true values. **d**, Type I error rate (for δ_{21}) was still well-controlled if the stratification effects could be adjusted. In the simulations, $\delta_{12} = 0.1$ and $\delta_{21} = 0.0$; $\pi_1 = \pi_2 = \pi_C = 1 \times 10^{-3}$; $h_1^2 = h_2^2 = 0.3$, $h_{C1}^2 = h_{C2}^2 = 0.1$ and $\rho_{C1,C2} = 0.1$.

To further address this concern about population stratification factors, we updated the method to allow for fixing the stratification parameters as 0.0. We tested the updated methods on $S_{1,2,C}$, $S_{2,C}$, and S_C scenarios (uni-directional causations) and the estimates were still unbiased and type I error

rate was controlled (Figure 2).

Figure 2. Estimates while fixing stratification parameters as zeros under $S_{1,2,C}$, $S_{2,C}$, and S_C scenarios. a, Estimates of δ_{12} and δ_{21} were nearly unbiased in all tested scenarios. **b,** Type I error rate (for δ_{21}) was controlled. In the simulations, $\delta_{12} = 0.1$ and $\delta_{21} = 0.0$; For $S_{1,2,C}$ scenario, $\pi_1 = \pi_2 = \pi_C = 1 \times 10^{-3}$, $h_1^2 = h_2^2 = 0.3$, $h_{C1}^2 = h_{C2}^2 = \rho_{C1,C2} = 0.1$; For $S_{2,C}$ scenario, $\pi_1 = h_1^2 = 0.0$, $\pi_2 = \pi_C = 1 \times 10^{-3}$, $h_2^2 = 0.3$, $h_{C1}^2 = h_{C2}^2 = \rho_{C1,C2} = 0.1$; For the S_C scenario, $\pi_1 = \pi_2 = h_1^2 = h_2^2 = 0.0$, $\pi_C = 1 \times 10^{-3}$, $h_{C1}^2 = 0.3$, $h_{C2}^2 = 0.4$, $\rho_{C1,C2} = 0.1$.

3. This response misses the point (also raised by reviewer 2). The "proof by simulation" show that you can detect causal effects of T2D liability on BMI. Our point was that such causal effects are implausible. If you have T2D then you might lose weight, but how it is plausible that T2D liability, in the general population, causes reduced BMI?

As the reviewer #2 has suggested, we also used a GWAS of T2D_adj_BMI (Mahajan A, et al. Nat Genet. 2018) and a GWAS of BMI (Yengo L, et al. Hum Mol Genet. 2018) to run the analysis. The reciprocal causations are still significant: the causal estimate from T2D to BMI is -0.13 (pval=5.8e-14) and the reverse causal estimate is 0.28 (pval=9.2e-21). We have updated the results in the manuscript.

4. This response does help. I still don't fully understand the S_C scenario. My intuition is that given the total SNP effects on X and Y there would be more than one solution for the two causal effects. But maybe this is reflected in the distribution of results for S_C , eg fig 3a the results for delta_12 look multimodal. I suspect the assumptions of normal effects may also have something to do with it.

Thanks for the comment. We admitted that this S_C scenario is the most challenging scenario for causal inference, this point has also been discussed and listed as a limitation in the main manuscript. To further prove this point, we showed the composite log-likelihood (CL) changes with respect to the two causal parameters under null causations (i.e. $\delta_{12} = \delta_{21} = 0.0$). The other parameters were fixed as true values (the stratification factors were fixed as the estimates by MRCI) when calculating CL space. As shown in Figure 3, for the $S_{1,2,C}$ scenario, the CL space shows an obvious maximum zone close to the true values of δ_{12} and δ_{21} . For the $S_{2,C}$ scenario, the maximum zone expands along the true value of δ_{12} , which is as expected since the Y_2 -specific component exists in this scenario. However, for the S_C scenario, the maximum CL could be achieved among many combinations of δ_{12} and δ_{21} , suggesting

more than one solution for the two causal effects. We cannot provide accurate causal effects under this S_C scenario. Instead, we proposed an averaged model to control the type I error rate at a reasonable level for this scenario.

Figure 3. The composite log-likelihood (CL) space for $S_{1,2,C}$, $S_{2,C}$, and S_C scenario under null causations. Representative scatterplots of each scenario were shown. The contour plots showed the CL space for each scenario by only varying values of δ_{12} and δ_{21} . In the simulations, $\delta_{12} = \delta_{21} = 0.0$; For $S_{1,2,C}$ scenario, $\pi_1 = \pi_2 = \pi_C = 1 \times 10^{-3}$, $h_1^2 = h_2^2 = 0.3$, $h_{C1}^2 = h_{C2}^2 = \rho_{C1,C2} = 0.1$; For $S_{2,C}$ scenario, $\pi_1 = h_1^2 = 0.0$, $\pi_2 = \pi_C = 1 \times 10^{-3}$, $h_2^2 = 0.3$, $h_{C1}^2 = h_{C2}^2 = \rho_{C1,C2} = 0.1$; For the S_C scenario, $\pi_1 = \pi_2 = h_1^2 = h_2^2 = 0.0$, $\pi_C = 1 \times 10^{-3}$, $h_{C1}^2 = 0.3$, $h_{C2}^2 = 0.4$, $\rho_{C1,C2} = 0.1$.

7. This is now ref 25, but the author and publisher are still missing. They should be provided in any format. It just needs editing in your referencing software.

We have added the author and publisher information as follows:

“Paxton, P., Hipp, J.R. & Marquart-Pyatt, S. Nonrecursive Models: Endogeneity, Reciprocal Relationships, and Feedback Loops. (SAGE Publications, Inc, Thousand Oaks, California, 2011).”

Reviewer #2 (Remarks to the Author: Overall significance):

The authors have done an immense amount of additional work. I agree with the authors that the utility of the proposed approach given potentially widespread pleiotropy effect. The revision of the manuscript has addressed all my previous comments.

We highly appreciate the reviewer's useful comments, which significantly improved our manuscript.

Reviewer #3 (Remarks to the Author: Overall significance):

This paper presents a novel method for MR estimation in the presence of correlated pleiotropy.

Reviewer #3 (Remarks to the Author: Impact):

This method will potentially be of use for other researchers in the field, particularly this method will provide another robust method that they can apply to use MR to estimate whether or not observed associations between traits are causal.

Reviewer #3 (Remarks to the Author: Strength of the claims):

The simulations are now convincing in showing when this method is likely to be applicable and useful. I am satisfied with the changes the authors have made and have no further comments on the paper.

Reviewer #3 (Remarks to the Author: Reproducibility):

The simulations and applied analysis are now sufficiently described for the reader to understand what has been done.

We highly appreciate the reviewer's useful comments, which significantly improved our manuscript.

REVIEWERS' COMMENTS:

Reviewer #1 (Remarks to the Author: Strength of the claims):

My previous comments have been well addressed. Thank you for this excellent addition to the MR toolkit.

One minor point: in the new population stratification simulations, "Caucasian" may not be an appropriate term, consider using "European ancestry".

We have revised the term as "European ancestry" accordingly. We really appreciate that the reviewer offered useful comments to improve our manuscript.